# Two-way coupling between ice flow and channelized subglacial drainage enhances modeled marine ice-sheet retreat

George Lu and Jonathan Kingslake

Lamont-Doherty Earth Observatory of Columbia University, Palisades, NY, USA

**Correspondence:** George Lu (george.lu@columbia.edu)

**Abstract.** Ice-sheet models used to predict sea-level rise often neglect subglacial hydrology. However, theory and observations suggest that ice flow and subglacial water flow are bidirectionally coupled: ice geometry affects hydraulic potential, hydraulic potential modulates basal shear stress via the basal water pressure, and ice flow advects the subglacial drainage system. This coupling could impact rates of ice mass change, but remains poorly understood. We develop a coupled ice–subglacial-
hydrology model to investigate the effects of coupling on the long-term evolution of marine-terminating ice sheets. We combine a one-dimensional channelized subglacial hydrology model with a depth-integrated marine-ice-sheet model, incorporating each component of the coupling listed above, yielding a set of differential equations that we solve using a finite-difference, implicit time-stepping approach. We conduct a series of experiments with this model, using either bidirectional or unidirectional coupling. These experiments generate profiles of channel cross-sectional area, channel flow rate, channel effective pressure, ice thickness, and ice velocity. We discuss how the profiles shape one another, resulting in the effective pressure reaching a local maximum in a region near the grounding line. We also describe the impact of bidirectional coupling on the transient retreat of ice sheets through a comparison of our coupled model with ice-flow models that have imposed static basal conditions. We find that including coupled subglacial hydrology leads to grounding-line retreat that is virtually absent when static basal conditions are assumed. This work highlights the role time-evolving subglacial drainage may have in ice-sheet change and informs efforts to include it in ice-sheet models. This work also supplies a physical basis for a commonly used parameterization which assumes that the subglacial water pressure is set by the bed's depth beneath the sea surface.

## 1 Introduction

The Ice-Sheet Model Intercomparison Project for CMIP6 (ISMIP6) predicts between 7.8 and 30.0 cm of sea-level-equivalent ice-mass loss from Antarctica by 2100 under the Representative Concentration Pathway 8.5 scenario (Seroussi et al., 2020). This wide range of predictions for the same forcing scenario is associated with various uncertainties within and differences between the ice-sheet models. A significant source of uncertainty is a limited understanding of how evolving subglacial hydrology could influence ice-sheet mass loss (De Fleurian et al., 2018; Flowers, 2015). A common approach to capturing the effect of hydrology on ice dynamics in ice-sheet models is to use measured surface velocities to invert for a spatially varying basal friction parameter (Arthern and Gudmundsson, 2010; Morlighem et al., 2013; Arthern et al., 2015; Lipscomb et al., 2021). This parameter encompasses all bed properties relevant for basal shear stress, including subglacial hydrology. Often,

this parameter remains static in time during simulations and does not evolve with changes in hydrology or basal water pressure (e.g., Lipscomb et al., 2021; Gudmundsson et al., 2019; Arthern and Williams, 2017). However, subglacial hydrology does evolve, and variations in basal water pressure have been linked to changes in ice dynamics (Bindschadler, 1983; Alley et al., 1994; Fountain and Walder, 1998; Clarke, 2005; Stearns et al., 2008).

Another way to represent subglacial hydrology in ice-sheet models is with simple parameterizations or with simple models (Kazmierczak et al., 2022; McArthur et al., 2023). These relate basal water pressure to the depth of the bed below sea level (e.g., Tsai et al., 2015) or the hydrological connectivity to the ocean (e.g., Leguy et al., 2014), or use simple hydrology models to estimate the depth of subglacial water (e.g., Le Brocq et al., 2009) or the till pressure (e.g., Bueler and Brown, 2009). However, ice-sheet sensitivity to subglacial hydrology varies based on the parameterization or model used (e.g., Kazmierczak

et al., 2022; Drew and Tarasov, 2023). Kazmierczak et al. (2022) showed that the range of sea-level-rise (SLR) predictions associated with these different parameterizations in a single ice-sheet model is similar in magnitude to the difference in SLR predictions between models used in ISMIP6 (Seroussi et al., 2020). Additionally, some of these simple parameterizations are likely not representative of the entire ice-sheet bed, and may only be valid near the grounding line (Leguy et al., 2014; Tsai et al., 2015; Kazmierczak et al., 2022).

We couple a physics-based subglacial hydrology model with an ice-sheet model, using a simple, one-dimensional approach. Our model avoids some of the common simple parameterizations for subglacial hydrology and allows the hydrology and the ice sheet to evolve in a coupled way. We use this model to examine how the coupling operates and investigate the assumption of holding basal friction parameters static in transient simulations.

Our work differs from previous coupled models of ice flow and hydrology by considering a marine-terminating setting, using

higher-order ice-flow physics than some models, and using a simpler hydrology model. For example, Kingslake and Ng (2013) and Arnold and Sharp (2002) coupled multi-element subglacial hydrology models, including channels and distributed linked cavities, to simplified ice-flow models, using an ice slab and the shallow ice approximation, respectively. In contrast, we use a higher-order ice-flow model including longitudinal stresses. Several other models (Pimentel and Flowers, 2010; Hewitt, 2013; Hoffman and Price, 2014; Gagliardini and Werder, 2018) couple similar multi-element hydrology systems to higher-order ice

flow models to study land-terminating glaciers. In contrast, we model marine-terminating glaciers and ice sheets. Finally, rather than using the multi-element hydrology models considered in other studies, we model a single, one-dimensional channel.

Our experiments include up to three points of coupling between the ice and hydrology models: [1] a sliding law that depends on effective pressure (which is the ice overburden pressure minus the basal water pressure) (Iken and Bindschadler, 1986), [2] a basal hydraulic gradient that depends on ice-sheet geometry (Fowler, 1999), and [3] a simple description of how the drainage

system is advected with the ice sheet as it flows (Drews et al., 2017). These points of coupling are illustrated by the solid lines in Fig. 1.

We consider marine-terminating ice due to its importance for future ice-sheet mass balance, and a channelized drainage system because observations and modeling suggest that there exist large, persistent channels underneath portions of the Antarctic Ice Sheet (Le Brocq et al., 2013; Drews et al., 2017; Dow et al., 2022). For simplicity, we do not model an adjacent distributed

drainage system, which would provide meltwater to the channel and influence basal sliding. This approach aids in interpreta-

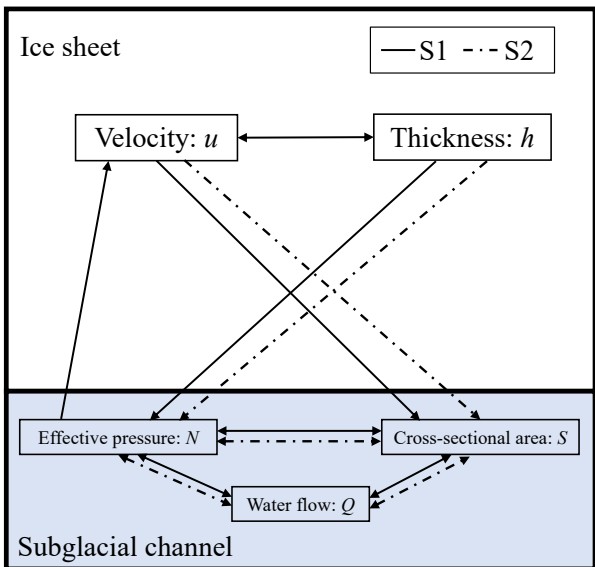

**Figure 1.** Points of coupling illustrated for our full coupled model (solid lines; S1), and a model with imposed ice geometry and velocity (dash-dot lines; S2).

tion of the physics of coupling and is supported by previous modelling suggesting that the pressures in channels and adjacent distributed cavities are closely coupled (Dow et al., 2022; Kingslake and Ng, 2013).

Section 2 describes our model and a set of steady-state and transient numerical experiments. Section 3 presents the results of these experiments. A key result is that the coupling between water and ice flow generates a region of relatively high effective pressure upstream of the grounding line in steady-state experiments. The transient experiments show the consequences of this region of high effective pressure for simulated retreat. Section 4 discusses and explains these results with the aid of a reduced version of the hydrology model and Section 5 draws conclusions about how these findings may apply to real systems.

## 2  Methods

Here, we describe the models of ice flow, basal sliding, and subglacial hydrology included in our coupled model. We also describe our numerical experiments. We first conduct a suite of steady-state experiments to understand the effects of parameter choices. We then examine a simulation in more detail to investigate the resulting effective pressure profiles. Next, we illustrate the importance of coupling for ice-sheet retreat through a set of transient experiments.

### 2.1  Ice dynamics

The ice-flow component of our model describes a two-dimensional, symmetric, shallow, marine ice sheet. The model uses an approximation usually referred to as the Shallow Shelf or Shelfy-Stream Approximation (SSA) (e.g., Schoof, 2007; Muszynski

and Birchfield, 1987; MacAyeal, 1989). It is depth-integrated and assumes rapid sliding, and its full derivation is described by Muszynski and Birchfield (1987). We use this model because it is one of the simplest models that accounts for longitudinal stresses, which are important near the grounding line. Mass and momentum balance are described by

$$\frac{\partial h}{\partial t} + \frac{\partial (hu)}{\partial x} = a, \tag{1}$$

$$\frac{\partial}{\partial x}\left[2A^{-1/n}h\left|\frac{\partial u}{\partial x}\right|^{1/n-1}\frac{\partial u}{\partial x}\right] - \tau_b - \rho_i gh\frac{\partial (h-b)}{\partial x} = 0, \tag{2}$$

where $h$ is the ice thickness, $u$ is the vertically uniform horizontal ice velocity, $x$ is distance from the ice divide, $t$ is time, $a$ is the ice-equivalent accumulation rate, $\tau_b$ is the basal shear stress, $\rho_i$ is the density of ice, $g$ is the acceleration due to gravity, $n$ is the Glen's law exponent, $A$ is a depth-averaged Glen's law coefficient, and $b$ is the bed depth below the sea surface (Schoof, 2007). For $b$ we use either a linear prograde slope or an overdeepened bed (Section 2.5). We use boundary conditions described by Schoof (2007). At the divide ($x = 0$) we impose symmetry, $\frac{\partial (h-b)}{\partial x} = 0$ and $u = 0$, and at the grounding line ($x = x_g$) we impose flotation $\rho_i h = \rho_w b$ and a stress balance based on coupling to a downstream ice shelf,

$$2A^{-1/n}h\left|\frac{\partial u}{\partial x}\right|^{1/n-1}\frac{\partial u}{\partial x} = \frac{1}{2}B\rho_i\left(1 - \frac{\rho_i}{\rho_w}\right)gh^2, \tag{3}$$

where $\rho_w$ is the density of water and $B$ is a nondimensional factor that represents the amount of ice-shelf buttressing being exerted on the grounded ice at the grounding line. This is used in transient experiments to perturb the system, following Brondex et al. (2017) and Drouet et al. (2013). This boundary stress condition accounts for flotation and the lack of basal shear stress at $x_g$, which is consistent with the assumption described later that effective pressure is zero at the grounding line. The mass balance (Eq. 1) dictates that thickness changes in the ice are due to the imbalance between the accumulation rate $a$ and the flux divergence. The momentum balance (Eq. 2) describes the balance between the longitudinal stress (first term), vertical shear stress (second term), and driving stress (third term). The hydrology influences the ice dynamics by modulating the basal shear stress (coupling point [1] from Section 1), which is represented by an effective-pressure-dependent sliding law.

## 2.2 Sliding laws

Different choices of sliding law for representing the basal shear stress $\tau_b$ yield different behaviors in ice-sheet models, especially near the grounding line (e.g., Brondex et al., 2017; Tsai et al., 2015; Barnes and Gudmundsson, 2022). Classically, models use a power law (also known as a Weertman law) to describe the relationship between $\tau_b$ and the sliding velocity (Weertman, 1957):

$$\tau_b = C_W u^m, \tag{4}$$

where $m$ is usually related to the exponent in Glen's flow law, $n$ (Paterson, 1994), and $C_W$ is a basal friction parameter that describes the bed properties. Typically, $m = \frac{1}{n} = \frac{1}{3}$ (e.g., Weertman, 1974; Schoof, 2007). Equation (4) does not explicitly consider the dependence that $\tau_b$ can have on the effective pressure, $N$, defined as the ice overburden pressure minus the subglacial water pressure.

The first $N$-dependent sliding law we adopt is an adjusted power law proposed by Budd et al. (1979):

$$\tau_b = C_B N^q u^m, \tag{5}$$

where $C_B$ is another friction parameter with the subscript used to distinguish it from $C_W$, and $q$ is a positive constant. Present models with this style of sliding law (which we refer to as a Budd sliding law) typically use a value of $q = 1$ (e.g., Brondex et al., 2017). One drawback of this sliding law is that it allows for arbitrarily high shear stresses, which is unphysical (Schoof, 2005; Brondex et al., 2017).

The second $N$-dependent sliding law we adopt is a regularized-Coulomb law (Helanow et al., 2021; Schoof, 2005):

$$\tau_b = C_C N \left( \frac{u}{u + A_s C_C^n N^n} \right)^{1/n}, \tag{6}$$

where $A_s$ and $C_C$ are two additional parameters describing bed properties Helanow et al. (2021). Sliding laws in this form have been used to represent sliding with cavitation on hard beds and sliding over deformable glacial till (e.g., Schoof, 2005; Zoet and Iverson, 2020; Helanow et al., 2021). In the limit of high $N$, this law behaves like a power law, Eq. (4). In the limit of low $N$ (typical near the grounding line), Eq. (6) reduces to a Coulomb plastic sliding law, $\tau_b \approx C_C N$, and does not have a strong dependence on ice velocity. Neither $N$-dependent sliding laws apply in the case when $N0$, so we avoid that scenario in simulations.

Our one-dimensional ice model can be considered to represent a narrow region of an ice sheet, narrow enough that ice stresses and the effective pressure at the ice-bed interface do not vary in the across-flow direction. Furthermore, we assume that a subglacial channel carved into the ice base exists at the ice-bed interface, is aligned with ice flow, and extends in the along-flow direction from the ice divide to the grounding line. We also assume that the basal effective pressure in this narrow region is equal to the effective pressure in the channel.

## 2.3 Subglacial hydrology

Our hydrology model describes the evolution of the subglacial channel and the balances of mass, momentum, and energy in the channel, respectively, as follows:

$$\frac{\partial S}{\partial t_h} = \frac{m}{\rho_i} - K_0 S N^3 - u \frac{\partial S}{\partial x}, \tag{7}$$

$$\frac{\partial S}{\partial t_h} + \frac{\partial Q}{\partial x} = \frac{m}{\rho_w} + M, \tag{8}$$

$$\psi + \frac{\partial N}{\partial x} = f \rho_w g \frac{Q|Q|}{S^{8/3}}, \tag{9}$$

$$mL = Q \left( \psi + \frac{\partial N}{\partial x} \right). \tag{10}$$

$S$ is the channel cross-sectional area, $N$ is the effective pressure, $Q$ is the channel discharge, $M$ is a constant and uniform supply term, $m$ is the melt rate, $t_h$ is the time for the hydrology system (which we distinguish from $t$ for convenience when nondimensionalizing later), $f$ is a hydraulic friction factor, $K_0$ is an ice flow parameter, and $\psi$ is the basic hydraulic gradient.

The closure term in Eq. (7) ($\frac{m}{\rho_i} - K_0 S N^3$) assumes a circular channel geometry. We assume that $M$ comes directly from the subglacial environment rather than from the ice surface.

The basic hydraulic gradient $\psi$ represents coupling point [2] (Section 1). It is the hydraulic gradient that would exist if the water pressure in the channel were equal to the ice overburden pressure $p_i$, which is approximated as $\rho_i g h$:

$$\psi = \rho_w g \frac{\partial b}{\partial x} - \rho_i g \frac{\partial h}{\partial x}. \tag{11}$$

We impose $N = 0$ at the grounding line because the water pressure under the ice is expected to approximately equal the ice overburden pressure in this location. We also force a channel discharge boundary condition at the divide, where we maintain a constant small influx $Q_{\text{in}}$ (effectively a Neumann boundary condition on $N$). We assume that the channel extends to the divide. To avoid this simplification causing the channel to become unrealistically large near the divide, we impose a very small $Q_{\text{in}}$. The boundary condition on $S$ at the divide is $\frac{\partial S}{\partial x} = 0$.

Our hydrology model is based on the modification by Fowler (1999) of the model for a one-dimensional subglacial channel from Nye (1976), with the addition of an ice advection term $u \frac{\partial S}{\partial x}$ in Eq. (7), following Drews et al. (2017). Equation (7) shows how the channel cross-sectional-area $S$ grows with melt $\frac{m}{\rho_i}$ and closes with ice creep $K_0 S N^3$. The advection term in Eq. (7) represents coupling point [3] (Section 1). It captures how the sliding of basal ice over the bed moves the roof of the channel. This term is required for a steady state to be reached, as we will explain in our results. Equation (8) balances the flux

associated with evolution of the channel cross-sectional area, flux divergence along the channel, and water gained through melt and additional water sources represented by $M$. Equation (9) uses Manning's equation (e.g., Chow, 1959; Röthlisberger, 1972) to describe the pressure gradient necessary to drive a flux $Q$ through a channel with a cross-sectional area $S$. Equation (10) details the assumption that all the energy dissipated by the turbulently flowing water is used locally to melt the channel walls.

## 2.4 Nondimensionalization and Numerics

To aid in numerically solving these equations and to understand the scales of each term, we nondimensionalize them (see Appendix A for the full procedure). There are six unknowns ($Q$, $N$, $S$, $h$, $u$, $m$) and six equations, but for simplicity we eliminate $m$ between Eq. (7), (8) and (10), reducing this to five unknowns and five equations. The dimensionless forms of the model equations are:

$$\frac{\partial S'}{\partial t'_h} = \frac{|Q'|^3}{S'^{8/3}} - S'N'^3 - \beta u' \frac{\partial S'}{\partial x'}, \tag{12}$$

$$\frac{\partial Q'}{\partial x'} = \epsilon(r-1)\frac{|Q'|^3}{S'^{8/3}} + \epsilon\left(S'N'^3 + \beta u' \frac{\partial S'}{\partial x'}\right) + M', \tag{13}$$

$$\frac{\partial N'}{\partial x'} = \frac{1}{\delta}\left(\frac{Q'|Q'|}{S'^{8/3}} - \psi'\right), \tag{14}$$

$$\frac{\partial h'}{\partial t'} + \frac{\partial(h'u')}{\partial x'} = a', \tag{15}$$

$$\alpha\frac{\partial}{\partial x'}\left[h'\left|\frac{\partial u'}{\partial x'}\right|^{1/n-1}\frac{\partial u'}{\partial x'}\right] - \tau'_b - h'\frac{\partial(h'-b')}{\partial x'} = 0, \tag{16}$$

and the associated parameters are: $\beta \equiv \frac{t_{h0}}{t_0}$, $\epsilon \equiv \frac{x_0 m_0}{Q_0 \rho_i}$, $r \equiv \rho_i/\rho_w$, $\delta \equiv \frac{N_0}{x_0 \psi_0}$, and $\alpha \equiv \frac{2u_0^{1/2}}{\rho_i g A^{1/n} h_{g0} x_0^{1/n}}$.

$\tau_b'$ depends on the sliding law choice. Nondimensionalizing Eq. (5) gives

$$\tau_b' = N' u'^{1/n}, \tag{17}$$

while nondimensionalizing Eq. (6) gives

$$\tau_b' = \gamma N' \left( \frac{u'}{u' + N'^n} \right)^{1/n}, \tag{18}$$

with $\gamma \equiv \frac{C N_0 x_0}{\rho_i g h_{g0}^2}$. Primes indicate dimensionless variables which can be returned to their dimensional forms by multiplying
with their scale (e.g., $x = x_0 x'$). For clarity, we drop the primes for the remainder of the analysis; unless otherwise stated, all variables referred to are their dimensionless forms. Constants for the experiments detailed in the subsequent sections are found in Table 2, while the resulting parameter and scale values are found in Table 3.

    Regardless of the sliding law used we find that $\beta \ll 1$ (Table 3), because the time scale for the hydrology, $t_{h0}$, is about four orders of magnitude smaller than the time scale for the ice sheet, $t_0$; $t_{h0}$ is on the order of months while $t_0$ is on the order of
millennia. We interpret this as the hydrology being in a "pseudo-steady" state, meaning that it quickly reaches a steady state for whatever conditions the ice imposes as the ice evolves. We exploit this to simplify the numerical solution (see below) of the hydrology equation, by setting the time derivative in Eq. (12) to zero. $\beta \ll 1$ also implies that the ice advection term in Eq. (12) has minimal effect. Regardless, we retain that term in anticipation of it becoming important in cases when $\frac{\partial S}{\partial x}$ is large. Another insight stemming from these parameter values is that $\alpha \ll \gamma$ and $\alpha \ll 1$, which implies, for the regularized Coulomb and Budd
cases, that the basal shear stress has a much larger impact on the ice-sheet force balance than the longitudinal stresses in Eq. (16). Regardless, to better represent the dynamics near the grounding line where the basal shear stress vanishes, we retain the longitudinal stress term.

    To solve these equations, we apply the finite-difference approximation following Schoof (2007). The model domain and variables are descritized into one-dimensional grids. The ice velocity and thickness grids are staggered and split into two
segments, one with coarse resolution and another with fine resolution near the grounding line. This split-grid approach is to preserve computational efficiency while properly resolving the stresses near the grounding line. The different grids used in our experiments are described in Table 1. To account for a moving grounding line, we apply a coordinate transformation (Appendix B) that allows for the "stretching" of the grids (Schoof, 2007). The grounding-line position is implicitly determined at each time step by imposing the flotation condition. For the hydrology equations, we use a uniform grid at an intermediate resolution.
The hydrology equations also undergo the same coordinate stretching as the ice equations. In each iterative step, the effective pressure is linearly interpolated onto the velocity grid while both the velocity and ice thickness are linearly interpolated onto the hydrology grid. Spatial derivatives are approximated with centered differences, unless they are at the boundary and not addressed by a boundary condition, in which case we use one-sided differences. Time derivatives are solved with a backwards Euler method. The system of nonlinear discrete equations is solved iteratively with MATLAB's *fsolve* function. This solution
method is based on code from Robel (2021), which was used to solve the model described by Schoof (2007).

**Table 1.** Experiments

| name | constant ice shape/velocity (Y/N) | constant hydrology (Y/N) | ice grid points | hydrology grid points | transience (Y/N) |
|---|---|---|---|---|---|
| sensitivity | N | N | coarse (85%): 100<br>fine (15%): 600 | 1000 | N |
| S1 | N | N | 3000 | 3000 | N |
| S2 | Y | N | 3000 | 3000 | N |
| T1 | N | N | coarse (85%): 100<br>fine (15%): 600 | 1000 | Y |
| T2 | N | Y | coarse (85%): 100<br>fine (15%): 600 | 1000 | Y |

## 2.5 Experimental design

Table 1 summarizes our suite of experiments. As detailed below, we conduct a series of steady-state simulations, followed by a series of transient simulations in which the ice evolves transiently and the drainage system evolves, but is assumed to be in a pseudo-steady-state.

### 2.5.1 Steady-state experiments

We run a suite of steady-state experiments across a range of parameters to examine model sensitivity. We conduct a separate sensitivity analysis using each of our two sliding laws. We vary four parameters while using the regularized Coulomb law: the accumulation rate $a$, the meltwater supply $M$, the ice stiffness $A$, and the sliding law coefficient $C_C$. For experiments with the Budd law, we vary three parameters: $a$, $M$, and $A$. We use four different values for each parameter. We limited the sensitivity analysis to these parameters and four different values per parameter to reduce computational time and to aid in visualization of the results. We selected the parameters that alter $\alpha$ and $\gamma$ in different ways. For example, raising $C_B$ and raising $A$ both reduce $\alpha$, so we only examine sensitivity to $A$ for simplicity. The sampled values for these parameters are found in Table 2. The range in $a$ encompasses realistic accumulation rates (Kaspari et al., 2004; Bodart et al., 2023). The range in $M$ was derived from estimating the amount of additional meltwater needed to obtain realistic water fluxes at the grounding line given a 100-km-long channel (Dow et al., 2022; Hager et al., 2022). The ranges in $A$ and $C_C$ were selected to encompass values used in previous studies (Schoof, 2007; Pimentel et al., 2010; Helanow et al., 2021). In this suite of experiments (referred to as "sensitivity" in Table 1), we use a linear bed with a slope of 0.001, which slopes downwards in the direction of ice flow from a depth of 100 m below sea level at the divide ($x = 0$). We model an unbuttressed ice sheet, meaning that $B = 1$ for all our steady-state experiments. We impose $Q_0 = 1$ m$^3$ s$^{-1}$, $h_0 = 1000$ m, and $x_0 = 100$ km, and all other scales are derived from these values. For the ice grid, we use 100 points along 85% of the domain (the coarse grid) and resolve the remaining 15% with 600 points

(the fine grid). The hydrology grid has a uniform spacing with 1000 points. These grids were chosen to properly resolve the region near the grounding line while preserving computational efficiency.

Outside of this suite of steady-state, sensitivity experiments, we also run a set of experiments at a higher, uniform spatial resolution with a single parameter combination to examine the model behavior and the nature of the coupling in more detail. These experiments are referred to as S1 and S2 (Table 1), and are represented by the lines in Fig. 1. We use the median values from our sensitivity test parameter space, and the resulting parameter and scale values are detailed in Tables 2 and 3. We maintain the same bed slope described above. We use 3000 grid points for the thickness, velocity, and hydrology grids. We use this finer grid to validate our grid spacing choices from the larger suite of steady state experiments experiments.

S1 consists of two steady-state experiments solving the full coupled model. S1.Budd (S1.B) uses a Budd sliding law (Eq. 5), while S1.Coulomb (S1.C) uses a regularized Coulomb sliding law (Eq. 6). We follow this nomenclature for the remaining experiments that use different sliding laws in our full coupled model. Note that although different sliding laws are used, the subglacial drainage system and the ice-flow components of the model are coupled in the same way in S1.B and S1.C.

S2 describes a simplified scenario with an imposed ice geometry and velocity. We impose an ice sheet with a quadratic shape, with its thickness described by

$$h = 1400 \times \left( \sqrt{\frac{2 \times 10^5 - x}{2 \times 10^5}} + 0.3363 \right) \text{m}, \tag{19}$$

where $x \in [0, 2 \times 10^5]$ m. The grounding line is defined to be 200 km away from the divide, as that is within the range of our S1.B and S1.C results. The 0.3363 factor is to make sure that the ice thickness at the grounding line equals the flotation thickness. This scenario also imposes a uniform velocity of about 31.5 m yr$^{-1}$, which is near the mean velocities from Experiments S1. These velocity and thickness values were selected to be the same order of magnitude as the results from the coupled experiments. We hold these velocity and thickness values constant and use them to solve only the hydrology equations for a steady state. Employing this one-directional forcing in this experiment will help us understand how ice geometry drives the hydrology. As illustrated in Fig. 1, imposing this ice geometry and velocity removes the point of coupling where the effective pressure modulates the basal shear stress.

### 2.5.2 Transient experiments

We also perform two suites of of transient experiments with the coupled model, which we call T1 and T2. Like the experiments in S1, this set of experiments uses both a Budd sliding law (T1.B, T2.B) and a regularized Coulomb sliding law (T1.C, T2.C). These experiments can be considered an extension of those by Brondex et al. (2017), who run a series of transient experiments using an SSA model with varying sliding law choices. Like Brondex et al. (2017), we introduce a nondimensional buttressing factor in Eq. (3) and set it to a value of 0.4 to determine an initial steady state for the ice sheet, before perturbing the model by increasing this factor, simulating a reduction in ice-shelf buttressing.

**Table 2.** Constants used in model experiments S1, S2, T1, and T2

| constant | description | default value | sensitivity range |
|---|---|---|---|
| $A$ | depth-averaged Glen's law coefficient* | $1.38 \times 10^{-25}$ s$^{-1}$Pa$^{-3}$ | $[3.9, 8.6, 19, 42] \times 10^{-26}$ s$^{-1}$Pa$^{-3}$ |
| $A_s$ | Coulomb law ice rheology and bed morphology coefficient | $2.26 \times 10^{-21}$ m s$^{-1}$Pa$^{-3}$ | |
| $a$ | accumulation rate | 0.3 m yr$^{-1}$ | [0.1, 0.23, 0.37, 0.5] m yr$^{-1}$ |
| $C_C$ | Coulomb law fitting coefficient* | 0.3 | [0.1, 0.23, 0.37, 0.5] |
| $C_B$ | Budd law coefficient | 7.624 m$^{-1/3}$ s$^{1/3}$ | |
| $f$ | channel friction factor | 0.07 m$^{-2/3}$ s$^2$ | |
| $g$ | gravitational acceleration | 9.81 m s$^{-2}$ | |
| $K_0$ | flow parameter for inward ice deformation | $1 \times 10^{-24}$ s$^{-1}$Pa$^{-3}$ | |
| $L$ | latent heat of fusion of water | $3.3 \times 10^5$ J kg$^{-1}$ | |
| $M$ | additional water supply along channel* | $1.31 \times 10^{-4}$ m$^2$ s$^{-1}$ | $[1, 4.64, 21.5, 100] \times 10^{-5}$ m$^2$ s$^{-1}$ |
| $n$ | Glen's law exponent | 3 | |
| $Q_{\text{in}}$ | influx at divide | 0.001 m$^3$ s$^{-1}$ | |
| $\rho_i$ | density of ice | 917 kg m$^{-3}$ | |
| $\rho_w$ | density of water | 1028 kg m$^{-3}$ | |

*Note that for the transient experiments, we use $A = 1.5 \times 10^{-25}$ s$^{-1}$Pa$^{-3}$, $C_C = 0.2$, and $M = 1 \times 10^{-5}$ m$^2$ s$^{-1}$.

For all transient experiments, we run the model to a steady state on a bed topography with an idealized sill and overdeepening from (Schoof, 2007):

$$b = - \left[ 729 - 2184.8 \times \left( \frac{x}{750\text{km}} \right)^2 + 1031.72 \times \left( \frac{x}{750\text{km}} \right)^4 - 151.72 \times \left( \frac{x}{750\text{km}} \right)^6 \right] \text{m}. \tag{20}$$

Parameter values used in the transient experiments were selected to generate an initial steady-state grounding-line position that is downstream of the over-deepening. Specifically, we use $A = 1.5 \times 10^{-25}$ s$^{-1}$Pa$^{-3}$, $C_C = 0.2$, and $M = 1 \times 10^{-5}$ m$^2$ s$^{-1}$ (Table 2). After a steady state is found, to perturb the model, we reduce ice-shelf buttressing effects by increasing the nondimensional buttressing factor ($B$ in Eq. 3) from 0.4 to 1 (meaning fully unbuttressed) linearly over the first 10 years of the simulation. This gradual reduction in $B$ ensures numerical stability and on the timescale of our experiments, does not effect our results significantly. These experiments run for 5000 years with yearly time steps and use the same grids used in the sensitivity experiments (Table 1).

In T1, we preserve all points of coupling in our model and observe the modelled grounding-line retreat after perturbing the buttressing factor. In T2, we examine the impact of assuming static (unvarying in time) basal conditions on modelled grounding line retreat. To do this, we eliminate the hydrology equations from the model and represent the static basal hydrology properties with an unchanging effective pressure profile that is used in the basal shear stress term in Eq. (2). To define $N$, we use values from the initial steady-state solutions of the coupled models in T1. This means that the initial ice-sheet profiles in T1 and T2 are identical. However, unlike experiments in T1, which allow the hydrology to evolve actively with the ice after the perturbation,

**Table 3.** Scales and parameter values computed using the constants in Table 2

| scale/parameter | expression | | value | |
| --- | --- | --- | --- | --- |
| | using Coulomb Law | using Budd Law | using Coulomb Law | using Budd Law |
| $Q_0$ | imposed channel discharge scale | | $1\ \mathrm{m^3\ s^{-1}}$ | |
| $h_0$ | imposed ice thickness scale | | $1000\ \mathrm{m}$ | |
| $x_0$ | imposed glacier length scale | | $100\ \mathrm{km}$ | |
| $\psi_0$ | $\psi_0 = \rho_w g \frac{h_0}{x_0}$ | | $101\ \mathrm{Pa\ m^{-1}}$ | |
| $S_0$ | $S_0 = \left(f x_0 \frac{Q_0^2}{h_0}\right)^{3/8}$ | | $2.07\ \mathrm{m^2}$ | |
| $m_0$ | $m_0 = \frac{\rho_w g}{L} \frac{Q_0 h_0}{x_0}$ | | $3.06 \times 10^{-4}\ \mathrm{kg\ m^{-1}\ s^{-1}}$ | |
| $t_{h0}$ | $t_{h0} = \frac{\rho_i L f^{3/8}}{\rho_w g} \left(\frac{x_0}{h_0}\right)^{11/8} Q_0^{-1/4}$ | | $6.2 \times 10^{6}\ \mathrm{s}$ | |
| $M_0$ | $\frac{Q_0}{x_0}$ | | $1 \times 10^{-5}\ \mathrm{m^2\ s^{-1}}$ | |
| $N_0$ | $N_0 = \left(\frac{\rho_w g}{K_0 \rho_i L f^{3/8}}\right)^{1/3} \left(\frac{h_0}{x_0}\right)^{11/24} Q_0^{1/12}$ | | $5.44 \times 10^{5}\ \mathrm{Pa}$ | |
| $u_0$ | $u_0 = \frac{\rho_w g A_s C_C^3}{K_0 \rho_i L f^{3/8}} \left(\frac{h_0}{x_0}\right)^{11/8} Q_0^{1/4}$ | $u_0 = \frac{\rho_i^4 g^2 K_0 L f^{3/8}}{\rho_w C_B^3} \left(\frac{h_0^{37}}{x_0^{13} Q_0^2}\right)^{1/8}$ | $9.80 \times 10^{-6}\ \mathrm{m\ s^{-1}}$ | $1.02 \times 10^{-5}\ \mathrm{m\ s^{-1}}$ |
| $t_0$ | $t_0 = \frac{K_0 \rho_i L f^{3/8}}{\rho_w g A_s C_C^3} \frac{x_0^{19/8}}{h_0^{11/8} Q_0^{1/4}}$ | $t_0 = \frac{\rho_w C_B^3}{\rho_i^4 g^2 K_0 L f^{3/8}} \frac{x_0^{21/8} Q_0^{1/4}}{h_0^{37/8}}$ | $1.02 \times 10^{10}\ \mathrm{s}$ | $9.78 \times 10^{9}\ \mathrm{s}$ |
| $a_0$ | $a_0 = \frac{\rho_w g A_s C_C^3}{K_0 \rho_i L f^{3/8}} \left(\frac{h_0}{x_0}\right)^{19/8} Q_0^{1/4}$ | $a_0 = \frac{\rho_i^4 g^2 K_0 L f^{3/8}}{\rho_w C_B^3} \frac{h_0^{45/8}}{x_0^{21/8} Q_0^{1/4}}$ | $9.80 \times 10^{-8}\ \mathrm{m^2\ s^{-1}}$ | $1.02 \times 10^{-7}\ \mathrm{m^2\ s^{-1}}$ |
| $\beta$ | $\beta \equiv \frac{A_s C_C^3}{K_0 x_0}$ | $\beta \equiv \frac{\rho_i^5 K_0 L^2 f^{3/4} g}{\rho_w^2 C_B^3} \frac{h_0^{13/4}}{x_0^{5/4} Q_0^{1/2}}$ | $6.10 \times 10^{-4}$ | $6.37 \times 10^{-4}$ |
| $\epsilon$ | $\epsilon \equiv \frac{\rho_w g h_0}{\rho_i L}$ | | 0.033 | |
| $r$ | $r \equiv \rho_i / \rho_w$ | | 0.892 | |
| $\delta$ | $\delta \equiv \left(K_0 \rho_i L f^{3/8} \rho_w^2 g^2\right)^{-1/3} \frac{Q_0^{1/12}}{x_0^{11/24} h_0^{13/24}}$ | | 0.065 | |
| $\alpha$ | $\alpha \equiv \frac{2 A_s^{1/3} C_C \rho_w^{1/3}}{\left(K_0 L A \rho_i^4 g^2\right)^{1/3} f^{1/8}} \frac{Q_0^{1/12}}{x_0^{19/24} h_0^{13/24}}$ | $\alpha \equiv \frac{2 f^{1/8}}{C_B} \left(\frac{\rho_i K_0 L}{g A \rho_w}\right)^{1/3} \frac{h_0^{13/24}}{x_0^{7/8} Q_0^{1/12}}$ | 0.0198 | 0.0201 |
| $\gamma$ | $\gamma \equiv C_C \left(\frac{\rho_w}{K_0 \rho_i^4 g^2 L f^{3/8}}\right)^{1/3} \frac{x_0^{13/24} Q_0^{1/12}}{h_0^{37/24}}$ | N/A | 1.81 | N/A |

experiments in T2 hold the effective pressure profile static in time. This emulates the approach, common in ice-sheet modelling, of deriving bed parameters with inverse methods and keeping them constant in time. Note that we still use Eq. (3) as a boundary condition on $u$, so the assumption of $N = 0$ inherent in that expression remains.

## 3 Results

### 3.1 Steady-state experiments

Figure 2 shows the five key steady-state model variables from S1.B and S1.C, which use the Budd and Coulomb sliding laws respectively, and our one-way forcing experiment S2, which use an imposed ice thickness $h$ and flow speed $u$.

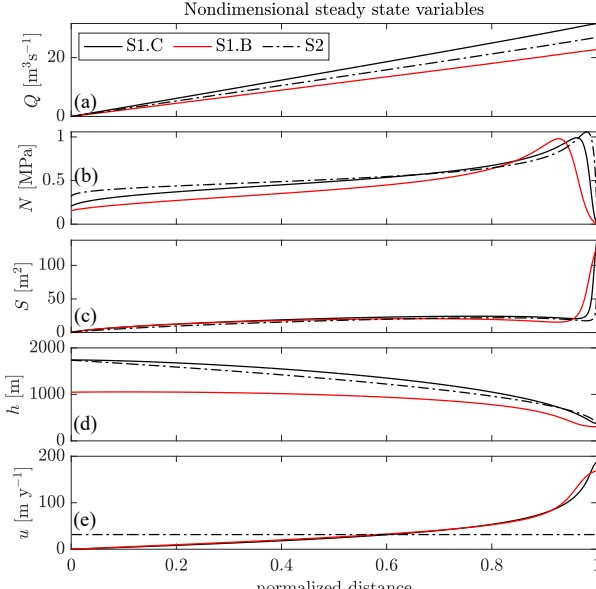

**Figure 2.** The steady-state profiles for: (a) channel discharge, (b) effective pressure, (c) channel cross-sectional area, (d) ice thickness, and (e) ice velocity. The results of the coupled simulations with the regularized Coulomb law (S1.C) are in black, the results from simulations using the Budd law (S1.B) are in red, and the results from the one-way coupled model with imposed thickness and velocity (S2) are in black dash-dotted curves.

Examining first the results from experiments S1.B and S1.C, $Q$ increases downstream (from left to right in Fig. 2a). The effective pressure $N$ gradually increases downstream before peaking (near $x = 0.93$ for S1.B, $x = 0.96$ for S1.C), then steeply decreasing towards the imposed $N = 0$ boundary condition at $x_g$. Between the peak in $N$ and $x_g$, $N$ passes an inflection point, where its curvature changes sign from negative upstream to positive downstream, and exhibits a slight taper. This means that the most negative effective pressure gradient is located upstream of the grounding line. The channel cross-sectional area $S$ gradually increases downstream from the divide before reaching a maximum, then decreasing downstream and reaching a minimum near the effective pressure peak, and finally steeply increasing towards $x_g$. $S$ also exhibits an inflection point very close to $x_g$ ($x = 0.985$ in S1.B and $x = 0.995$ in S1.C), then tapers towards $x_g$. The ice thickness $h$ decreases monotonically with $x$, while its gradient increases in magnitude for most of the domain until near the grounding line, where it passes an inflection point (near $x = 0.93$ for S1.B, $x = 0.98$ for S1.C) and the gradient begins to decrease in magnitude, approaching zero at $x_g$. The ice velocity $u$ increases downstream. Like the gradient in $h$, the gradient in $u$ increases in magnitude before reaching an inflection point near the thickness inflection points, then approaching zero at $x_g$. In Figure 2, these points of inflection are clearer in the S1.B profile.

The results from S2, with the imposed quadratic ice geometry, follow similar trends to S1: $Q$ increases monotonically, $S$ increases gradually with $x$ in the region $x < 0.5$, then decreases slightly before increasing steeply towards $x_g$, and $N$ increases

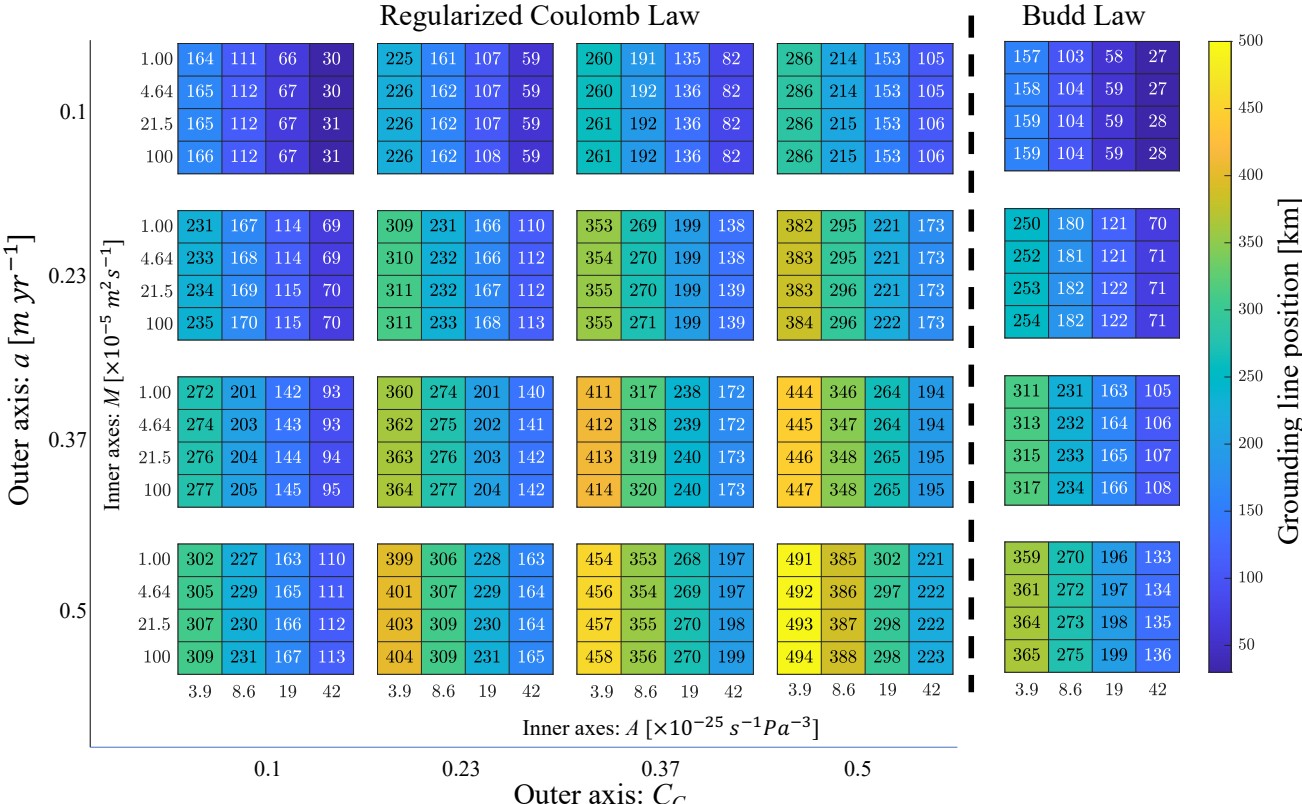

**Figure 3.** The sensitivity of the coupled model's steady-state grounding line position to the depth-averaged flow law constant $A$ (inner x-axis), the additional water source term $M$ (inner y-axis), and the accumulation rate $a$ (outer y-axis). Each column of plots corresponds to a different value of $C_C$ for the model using a regularized Coulomb law (outer x-axis), except for the rightmost column which contains results from experiments that used a Budd law.

downstream from the divide before peaking then decreasing rapidly to zero at $x_g$. The main difference is that these variables do
not have a point of inflection immediately upstream of the grounding line as seen in the results from the coupled model (S1).
This highlights how the ice surface topography controls the hydrology: the ice geometry in the coupled experiments exhibits
an inflection point and this is reflected in the hydrology profiles, whereas the quadratic ice geometry (Eq. 19) does not, and
hence when this is used to drive the hydrological part of the model, the results also lack these inflection points.

Figure 3 shows the sensitivity of steady-state grounding line positions with respect to variations in $A$, $a$, $M$, and (for models
using a Coulomb law only) $C_C$ for our coupled models. Grounding line position increases as $A$ decreases, or as $a$, $M$, or $C_C$
increase. The variations in $A$, $a$, and $C_C$ all result in sizeable differences in grounding-line position, whereas variations in $M$
have minimal effect. We note that the grounding-line position simulated using the low-resolution split grid in the sensitivity
experiments differs from those simulated using the high-resolution grid by $< 0.7\%$ for both S1.B and S1.C.

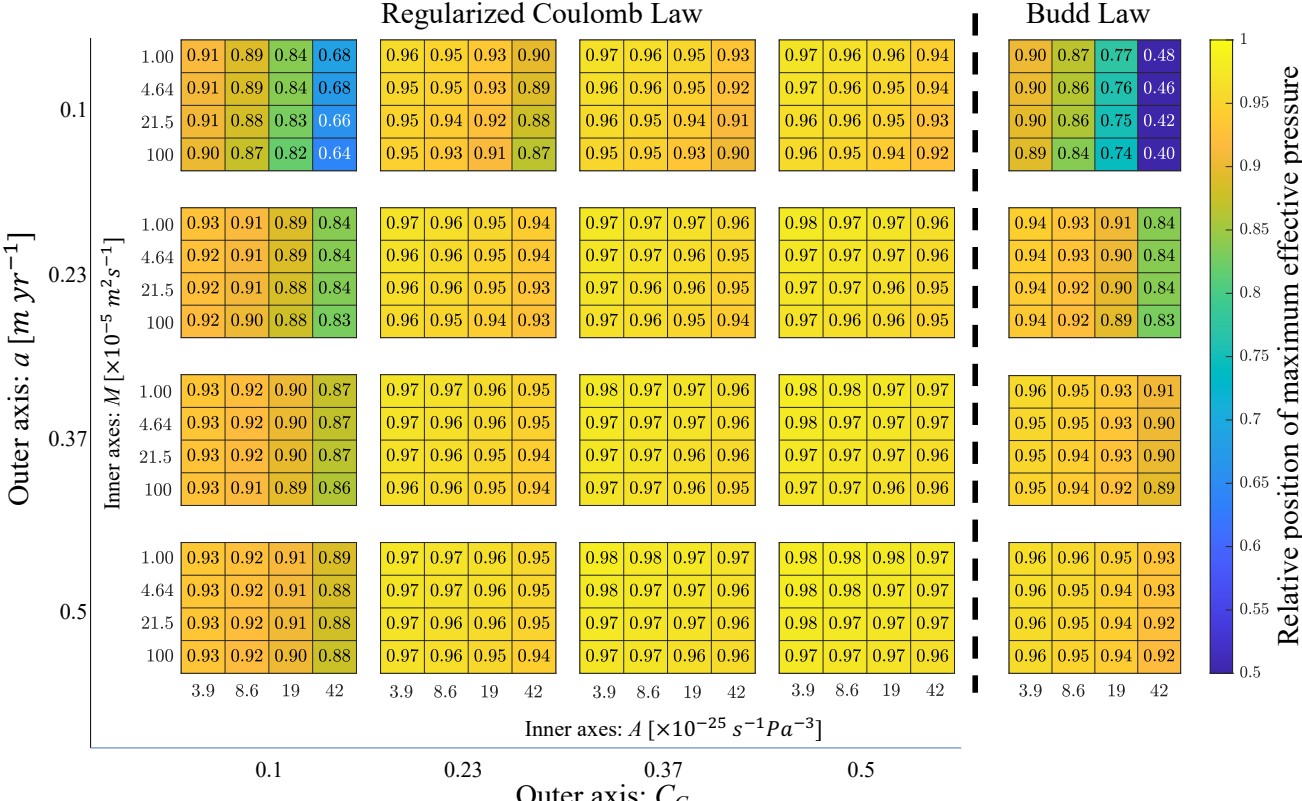

**Figure 4.** The sensitivity of the coupled model's relative location (0 is the divide, 1 is the grounding line) of peak effective pressure to the depth-averaged flow law constant $A$ (inner x-axis), the additional water source term $M$ (inner y-axis), and the accumulation rate $a$ (outer y-axis). Each column of plots corresponds to a different value of $C_C$ for the model using a regularized Coulomb law (outer x-axis), except for the rightmost column which contains results from experiments that used a Budd law.

In our steady-state experiments we noted a peak in effective pressure near the grounding line (c.f. Figure 2b). As we will

discuss in the next section, this peak in effective pressure plays a role in our transient experiments, so we are interested if this feature is persistent across parameter variations. Figure 4 shows the sensitivity of the location of maximum effective pressure with respect to the same parameter variations explored in Figure 3. The peak moves further from the grounding line as $A$ or $M$ increases, or as $a$ or $C_C$ decrease. $A$ and $a$ more significantly impact the location of the peak in effective pressure. We also see that the peak in effective pressure is consistently near the grounding line, with the exception of in the cases of

low accumulation, high $A$, and low $C_C$, where the peak is further upstream. Comparing these positions with their respective grounding-line locations shows that these are instances where the ice sheet is very small, and likely exhibiting plug flow.

While our focus is not the intercomparison of models with different sliding laws (e.g., Brondex et al., 2017), we note that sliding law choice does not change the general trends in model sensitivity to $A$, $a$, or $M$. Regardless of the sliding

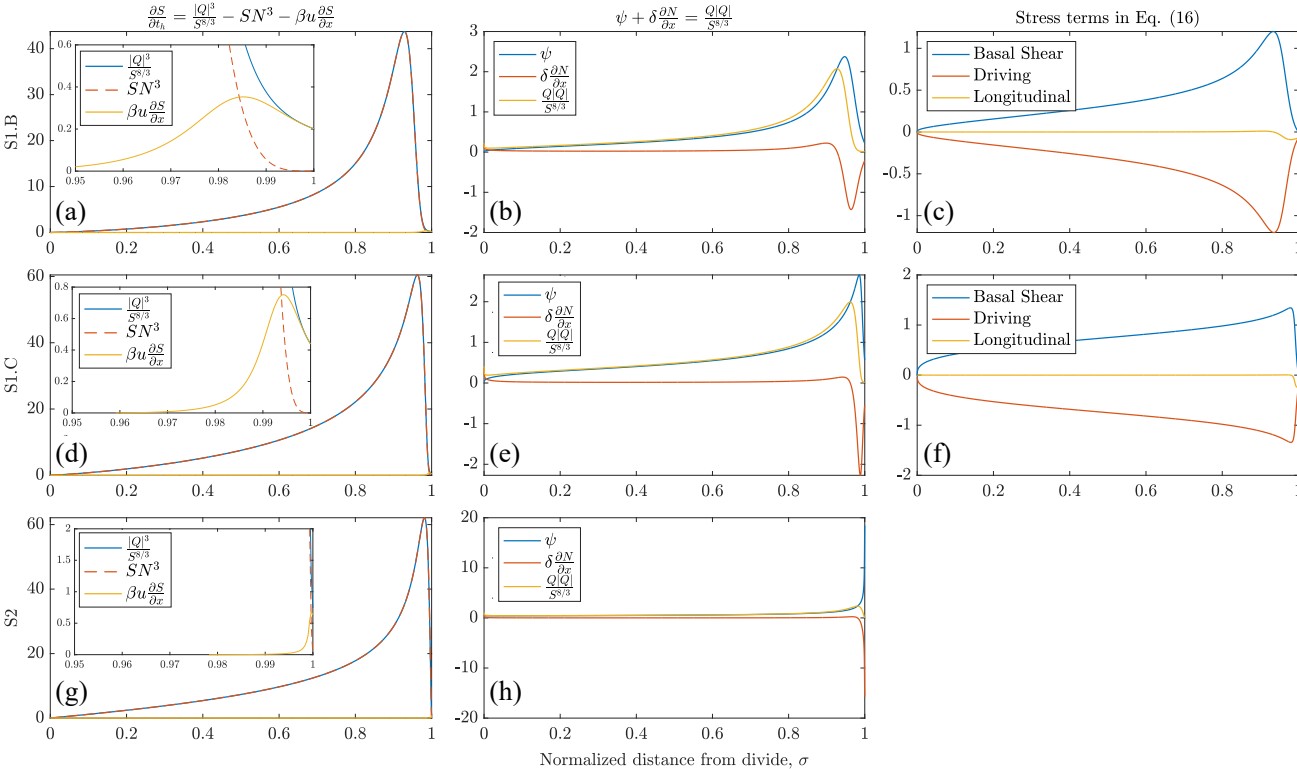

**Figure 5.** Term-by-term plots of the steady-state solution to the coupled model (S1). The terms in the equation for channel evolution (left column: a, d, g), in the equation for conservation of momentum in the channel (middle column: b, e, h), and in the equation for conservation of momentum in the ice (right column: c, f) for the steady-state coupled model solution using a Budd sliding law (top row: a-c), the solution using a regularized Coulomb law (middle row: d-f), and the one-way forced solution using an imposed quadratic ice geometry (bottom row: g, h).

law or parameter values, the effective pressure tends to peak near the grounding line, which implies a steep drop to zero at
the grounding line. This steep negative gradient in $N$ plays an important role in our transient experiments, so we return to Experiments S1 and S2 to examine this feature further.

    For each of the three experiments (S1.B, S1.C and S2), Fig. 5 shows the nondimensional values of each term in Eqs. (12, 14, 16) to further illustrate how the ice and the hydrology influence one another. In the first column, we see the small but important effect of ice flow advecting the subglacial channel. We anticipated this term to be small, as $\beta \ll 1$. However, across the three
experiments, as $N$ approaches zero at the grounding line, creep closure ($SN^3$ in Eq. 12) also approaches zero, as illustrated by the red dashed curves in Fig. 5. In the absence of advection, this would need to be accommodated by a reduction in the melt term $\frac{|Q|^3}{S^{8/3}}$. Given that $Q > 0$, this would require $S$ to grow to infinity. We see from the steady-state profile in Fig. 2c that

$S$ grows large, but does not grow to infinity, and that is due to the advection term growing near the boundary, contributing to channel closure (because $\frac{\partial S}{\partial x} > 0$), and therefore helping to balance the non-zero melt (insets, Fig. 5).

The middle column of Fig. 5 explains the relationship between effective pressure and ice geometry. This is most clearly seen in the one-way experiment (S2; Fig. 5h). Consistent with $\delta \ll 1$, for most of the domain $\frac{Q|Q|}{S^{8/3}}$ follows $\psi$, which is set by the ice geometry and bed topography. However, as we approach $x_g$, $\frac{Q|Q|}{S^{8/3}}$ and $\psi$ diverge; as we approach the $N = 0$ boundary condition, $S$ grows large and $\frac{Q|Q|}{S^{8/3}}$ approaches zero. In contrast, $h$ growing steeper near the grounding line causes $\psi$ to grow large. To balance this mismatch between the gradient required to drive flow $(\frac{Q|Q|}{S^{8/3}})$ and the hydraulic gradient supplied by the

ice geometry $(\psi)$, $\delta\frac{\partial N}{\partial x}$ grows large and negative. This explains the steep decrease in $N$ just upstream of the grounding line (Fig. 5h).

The same relationships hold for the results from the coupled experiments (S1), with some modification related to the inflection points induced by the ice geometry, discussed above. In both S1 experiments, the $\frac{Q|Q|}{S^{8/3}}$ drops below $\psi$ near the grounding line, and the difference is made up by a negative $\delta\frac{\partial N}{\partial x}$ (just as in S2). In contrast to the results from S2, an inflection point

in the ice thickness influences the hydrology; $\psi$ no longer grows to a maximum at $x_g$, as it does with the imposed quadratic geometry. Instead, $\psi$ reaches a maximum upstream of the grounding line, causing the effective pressure gradient to reach a minimum slightly upstream of the grounding line (c.f. the red curves in the middle column of Fig. 5 ). This minimum shows up as a point of inflection in the $N$ profiles for the S1 results that is not seen in the S2 results.

Regardless of the whether the ice and hydrology are coupled (S1) or if the hydrology is forced by an imposed ice thickness

and velocity (S2), $\delta\frac{\partial N}{\partial x}$ is negative near the grounding line in response to high $\psi$ in this region. This results in $N$ growing large as you move upstream from the grounding line.

Figures 5c and 5f show the steady-state stress balance from S1.B and S1.C. The basal shear stress profiles illustrate the effect the hydrology has on ice flow, as they mimic the $N$ profiles, particularly near the grounding line. For example, the basal shear stress is highest where the effective pressure is highest and both drop to zero at $x_g$. As anticipated (due to $\alpha \ll 1$), the

longitudinal stresses are small for the majority of the ice sheet. However, they increase near the grounding line where the basal shear stress drops to zero.

In summary, regardless of variation in parameters (Figs. 3 and 4), as long as the gradient in ice thickness grows in the downstream direction, the effective pressure grows to a maximum before dropping to zero at the grounding line. This steep negative gradient in $N$ means that regions of high effective pressure exist just upstream of the grounding line. This in turn

causes a region of higher basal shear stress. Next, we explore the consequences of this high basal shear stress region for grounding line retreat if basal properties are assumed static in time.

### 3.2   Transient Experiments

The solid curves in Fig. 6 show results from transient simulations using the coupled models with the Budd law (T1.B) and the regularized Coulomb law (T1.C). The ice sheet and hydrological system start in a steady state with the grounding line

near the top of the sill, found by solving the coupled model in a steady state using a buttressing parameter of 0.4 imposed on the longitudinal stress boundary condition (Eq. 3). The transient simulation starts with this parameter being increased to 1

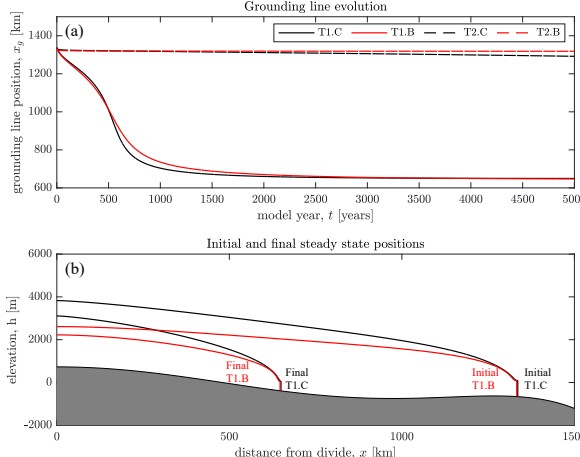

**Figure 6.** (a) The grounding-line position and (b) the initial and final ice-thickness profiles of the coupled model with a regularized Coulomb law (T1.C, black solid curves) and a Budd law (T1.B, red solid curves) over 5000 years of evolution after fully removing ice-shelf buttressing (by changing $B$ from 0.4 to 1) over 10 years. The dashed curves show the evolution of the static-$N$ experiment using a regularized Coulomb law (T2.C, black dashed curve) and a Budd law (T2.B, red dashed curve). The upper boundary of lower shaded region is the ice-sheet bed.

(corresponding to no buttressing), which triggers ice thinning and grounding-line retreat. The grounding line rapidly retreats past the overdeepening towards a steady state on the upstream prograde slope, a total retreat of about 678 km for T1.B. In T1.C, the grounding line also undergoes rapid retreat to a steady state on the prograde slope, moving almost 684 km. Looking closely near the grounding line of the T1 results in Fig. 7 (solid curves, bottom panels), we see the same peak in $N$ and points of inflection in $h$, $u$, and $N$ that we saw in the steady-state solutions on the linear bed. This peak in $N$ persists throughout the retreat over the prograde and reverse bed slopes. This suggests that the relationships between these variables discussed above in relation to the steady-state results also apply on this more complex bed topography and during retreat. Next we compare this coupled evolution with the evolution of an uncoupled model that uses a static effective pressure profile.

Figure 7 and the dashed curves in Figure 6a show results from T2.B and T2.C, where the effective pressure profiles are held static throughout the simulations at their initial values, derived from the coupled solution that provided the initial steady-state conditions. Under these conditions the grounding line, instead of retreating past the overdeepening, has only moves a relatively small distance upstream of its initial position. In the first fifty years of the simulations, the grounding line in T2.B moves approximately 12 km upstream of its starting position (Fig. 7a). In T2.C, the grounding line retreats only around 10 km in this fifty year period (note however that it does not approach a steady state until later in the simulation). This minimal retreat is in stark contrast to the coupled experiments (T1), which exhibit faster and larger magnitude grounding-line retreat.

We identify the following reasons for these contrasts. In the T2 experiments, the effective pressure is no longer near zero immediately upstream of the terminus during retreat. After the grounding line begins to retreat, the basal shear stress near the terminus grows. The increasing shear stress slows ice flow (c.f. dashed and solid curves in Figs. 7e,f) and prevents further

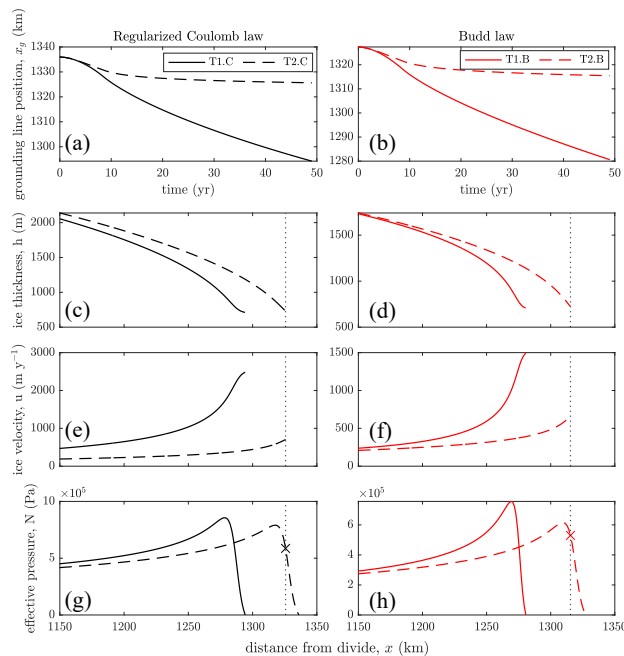

**Figure 7.** The grounding-line position over 50 years of evolution for the coupled experiment (T1, solid curves) and static-$N$ experiment (T2, dashed curves) using a (a) regularized Coulomb law (black curves) and (b) Budd law (red curves). The (c,d) ice thickness, (e,f) ice velocity, and (g,h) effective pressure profiles after 50 years of evolution. The vertical dotted lines show the position of the static-$N$ grounding lines after 50 years. For the static-$N$ effective-pressure profiles, we plot the initial steady-state profiles and the crosses indicate the effective pressure immediately upstream of the grounding line after 50 years.

grounding-line retreat. Further retreat results in even higher $N$ values, producing a negative feedback on the ice flow, as the basal shear stress increases further. The coupled model, on the other hand, allows $N$ immediately upstream of the grounding line to remain close to zero as the grounding line retreats; the hydrological system evolves in response to the changing ice geometry to keep $N$ (and therefore basal shear stress) low upstream of the grounding line, which in turn facilitates further retreat.

These differences between the T1 and T2 simulations can also be understood in terms of the ice stress balance, as follows. The removal of buttressing results in a reduction in longitudinal stress and an acceleration in ice flow near the grounding line. In T1, effective pressure is near zero immediately upstream of the grounding line so this acceleration does not lead to sufficiently increased basal shear stress to counteract the reduction in longitudinal stress. The acceleration leads to thinning and grounding-line retreat. In contrast, in Experiment T2, when $N$ does not change with time, grounding-line retreat corresponds to a higher $N$ near the boundary (and therefore higher basal shear stress) which grows to balance the reduction in longitudinal stress stress, preventing further acceleration and retreat.

## 4  Discussion

We have developed a model of subglacial hydrology that is bidirectionally coupled to ice flow. In all solutions to the model, the subglacial drainage system grows large in the region upstream of the grounding line, and an area of high effective pressure

forms immediately upstream of this region. This feature forms independent of whether we use a Budd or regularized Coulomb sliding law. A suite of sensitivity tests shows that the peak in $N$ occurs across a wide parameter space. Our hydrology-only experiment shows how imposed ice geometries and velocities can produce similar profiles of $N$, though our transient experiments, discussed later, highlight how two-way coupling (specifically between ice geometry, basal hydraulic gradient, subglacial-channel size, and effective pressure) is needed for realistic retreat. Our plotting of terms in Fig. 5 demonstrates the

roles different processes play in the effective-pressure profiles. Next, to further explore what occurs near the grounding line, we create a reduced model for the effective pressure, and show that it cannot apply at the grounding line. We discuss the behaviour of channel size at the grounding line, and comment on how a channel size increase feeds back on the effective pressure.

First, we reduce our coupled model by identifying parameters that are small in our dimensionless equations. $\delta$ is small, which means that the flow of water is primarily driven by the basic hydraulic gradient rather than the effective pressure gradient, at

least in locations where effective-pressure gradients are not large. $\epsilon$ is small, which means that the water flow gradient is primarily driven by additional water sources along the channel, rather than by melt or creep of the channel walls. As identified earlier, the timescale for ice dynamics is much greater than that of the hydrology, meaning that $\beta$ is small. Knowing that $\delta$, $\epsilon$, and $\beta$ are small, Eqs. (12-14) can be reduced to:

$$N = \frac{Q}{S^{11/9}}, \tag{21}$$

$$\frac{\partial Q}{\partial x} = M, \tag{22}$$

$$S = \frac{Q^{3/4}}{\psi^{3/8}}. \tag{23}$$

Combining Eqs. (21) and (23) gives

$$N = Q^{1/12} \psi^{11/24}, \tag{24}$$

which holds for most of the ice sheet far from the grounding line. Equation (22) indicates that $Q$ increases downstream.

Assuming that the bed slope is small ($\frac{\partial b}{\partial x} \ll 1$) and that the ice sheet gets steeper further from the divide, meaning that $-\frac{\partial h}{\partial x}$ grows downstream, Eq. (24) grows downstream from the divide. However, Eq. (24) does not satisfy $N = 0$ at the grounding line. Consequently, a separate near-field solution would be required to describe a boundary layer in $N$ near $x = x_g$. This requires a negative gradient in $N$ within the boundary layer such that the upstream $N$ represented by Eq. (24) will match with the $N = 0$ boundary condition.

The $N = 0$ boundary condition also impacts the channel cross-sectional area, $S$. As we approach the grounding line, $N$, and therefore the creep closure term $SN^3$ in Eq. (12), approach zero. This drop in $SN^3$ results in the melt-opening term in Eq. (12) dominating, causing the channel cross-sectional area to grow large near the terminus. The channel in this region therefore grows until the advection term $\beta u \frac{\partial u}{\partial x}$ grows large enough to counteract the melt opening. We note how the fact that Eq. (24)

only applies in regions far from the grounding line supports this interpretation; Eq. (24) shows that, for a channel with $Q \neq 0$, $N = 0$ requires $S$ to be infinite. This is not physical, but it is conceptually consistent with $S$ growing large near the grounding line until the advection term, which is neglected in the reduced model, starts to play a role. This result highlights the importance of the ice velocity advecting the drainage system, as it allows the coupled model to reach a steady state (see also Drews et al., 2017).

Another way of looking at this is that channel cross-sectional area $S$ growing large at the grounding line facilitates $N$ dropping to zero. Specifically, as discussed in Section 3, a large $S$ results in the right side of Eq. (9) becoming small, which results in the effective pressure gradient approximately equalling the negative of the basic hydraulic gradient, i.e. $\psi \approx -\frac{\partial N}{\partial x}$. Integrating this expression using the boundary condition $N(x_g) = 0$ yields

$$N \approx -\rho_w g b + \rho_i g h. \tag{25}$$

This expression is consistent with $N = 0$ at the grounding line, because the flotation condition is $\rho_i g h = \rho_w g b$. This expression, which is derived simply by assuming $S$ is large, thus provides the negative gradient required to bring $N$ to 0 from its far-field solution, as described by Eq. (24). Notably, this result is equivalent to assuming that the hydrology system is hydrostatically connected to the ocean, as done in previous modelling studies (e.g., Tsai et al., 2015; Brondex et al., 2017). The above discussion provides physical justification for this assumption and highlights how this simplification can be justified by the channel cross-sectional area growing large near the grounding line through ice-water coupling.

Consistent with previous work, we find that $h$ and $u$ pass inflection points in the region upstream of the grounding line, with the ice-thickness gradient tapering towards zero at the grounding line. Tsai et al. (2015) examined this feature using an ice-sheet model that assumed full hydrostatic connectivity with a Coulomb law (i.e. basal shear stress was assumed to be proportional to effective pressure near the grounding line). This assumption means that, like in our model, $N$ and $\tau_b$ vanished at the grounding line. Consistent with our description of the stress balance near the grounding line in our model, they showed that these inflection points arise due to the vanishing basal shear stress at the grounding line (Tsai et al., 2015).

One potential implication of our steady-state experiment results is that, in areas with channelized subglacial drainage and relatively steep ice thickness profiles, the region immediately upstream of the grounding line will experience high effective pressure and basal shear stress. Previous models are consistent with such a spatial distribution of effective pressure (e.g., Dow et al., 2018; De Fleurian et al., 2018; Hayden and Dow, 2023). For example, Hayden and Dow (2023) impose a realistic ice geometry on a two-dimensional multi-element subglacial-hydrology model, and their modelled effective pressures show the same abrupt increase in effective pressure immediately upstream of the grounding line that our model exhibits, regardless of whether the water flow is channelized or distributed. McCormack et al. (2022) inferred high basal shear stresses in the downstream portion of Thwaites Glacier and find lower basal shear stresses further upstream. While McCormack et al. (2022) discuss spatial variations in basal roughness (Schroeder et al., 2014) and in drainage-system configuration (Schroeder et al., 2013) as contributors to this spatial pattern, we propose that it could also be explained by coupling between ice geometry, basal hydraulic gradient, subglacial-channel size, and effective pressure. Future efforts to discriminate the control that hydrology has

on basal friction from other factors will be valuable, particularly considering that the hydrologically controlled component of basal shear can potentially change rapidly (e.g., Das et al., 2008; Joughin et al., 2013).

Temporal evolution is the focus of our transient experiments. In the steady-state simulations, the ice thickness and velocity control the hydrology more consequentially than the other way around; this is demonstrated by experiment S2, in which we imposed the ice thickness and velocity. The transient experiments, by contrast, highlight the role hydrology can have on the ice. The key takeaway from these experiments—designed to emulate the approach of inverting for bed properties and leaving them unchanged that is used in most larger-scale ice-sheet models—is that holding the hydrology static severely impacts ice flow near the grounding line. In our model, this results in a significant reduction in grounding-line retreat. The evolution of the hydrology system together with the ice facilitates faster, larger-magnitude retreat. This follows results from Brondex et al. (2017), who use velocity and shear-stress profiles determined using a regularized Coulomb law to invert for a sliding parameter $C_W$ in a Weertman sliding law that is held static. Their subsequent simulations with the Weertman sliding law result in minimal grounding-line retreat (Brondex et al., 2017). Inverting for $C_W$ and holding it static is similar to our approach of holding $N$ static. We confirm that this static assumption is what causes the lack of retreat, show that it also applies to Budd and regularized Coulomb sliding laws, and reveal the processes that control $N$ in the region upstream of the grounding line.

The stark difference in retreat accommodated by the coupled models compared to that of the static-$N$ models has implications for larger ice-sheet-modeling efforts. In many state-of-the-art ice-sheet models, it is common to invert for a spatially variable basal shear-stress or sliding-law parameter that encompasses all basal variables, subglacial hydrology included, and keep this static in time (e.g., Arthern and Gudmundsson, 2010; Morlighem et al., 2013; Arthern et al., 2015). However, these inversions are based on present-day measurements and holding the resulting bed properties constant does not account for future ice-sheet evolution (Arthern et al., 2015). Some large-scale ice-sheet models evolve basal conditions, but it is done through parameterizations (e.g., Leguy et al., 2021, 2014; Kazmierczak et al., 2022). Our work suggests a physical basis for one approach of assuming perfect hydraulic connectivity, but it is currently uncertain how good an approximation this is, particularly away from the region immediately upstream of the grounding line. This uncertainty emphasizes the need for efforts to better represent subglacial hydrology in ice sheet models.

Finally, although our coupled model includes the detailed physics of both a subglacial channel and a one-dimensional, marine-terminating ice sheet, it employs many simplifying assumptions. For the ice component, we neglect vertical velocities and vertical variability in horizontal velocities. We also neglect variation in ice temperature and lateral variability in the ice. This means that our model is applicable only to an ice sheet with significant basal sliding. Variations in temperature would result a nonuniform flow-law coefficient $A$ and alter ice dynamics, and variations at the ice-sheet base could result in ice freezing to the bed or in additional basal melt. For the hydrology component, we model only a single subglacial channel in pseudo-steady-state, and do not consider distributed or multi-channel systems. This simplification leads to another assumption that a region of the ice sheet large enough to control the ice dynamics has an effective pressure equal to the channel's effective pressure. Kingslake and Ng (2013)'s results suggests that to a first order, the effective pressure in a distributed linked cavity system connected to a channel follows that of the channel, but this will not be true far from the channel. Our findings therefore apply only to regions where channelized drainage systems dictate the water pressure. Additionally, we assume a uniform

and constant supply term $M$ and input term $Q_{in}$, when in reality the channel could be supplied by spatially and temporally varying sources. Related to this, we assumed a pseudo-steady-state in the hydrology component of the model in numerical experiments. This was motivated by a scaling analysis which used the properties of the coupled ice-hydrology system (in particular, $x_0$, $h_0$ and $Q_0$) to derive time scales of the hydrology system, rather than the time scale of external forcings. Therefore, the pseudo-steady-state assumption would not apply if the timescales of, for example, meltwater input to the system were shorter, perhaps due to fluctuations in the flux of surface meltwater reaching the bed. Finally, we assume that the channel is formed by incision upwards into the ice, whereas observations and models suggest that channels can also be formed by incision downwards into sediments (Ng, 2000; Livingstone et al., 2016). Including these processes in our model, along with sediment deposition (Drews et al., 2017), would significantly impact its behavior. These simplifications and assumptions make our model unlikely to quantitatively reproduce observations, capture seasonal changes in hydrology, or to be applicable in areas where drainage systems change their configuration between distributed and channelized over space and time. Despite these limitations, the model's simplicity allows a more complete understanding of the model's behavior than would be possible if a more comprehensive model were used. This simplicity has allowed us to qualitatively demonstrate, and understand in detail, some of the ways that active subglacial hydrology could impact long-term ice-sheet retreat.

## 5   Conclusions and Outlook

We have developed a model that uses a novel combination of physical coupling points between a marine-terminating ice sheet and a subglacial channel. We allow the ice-sheet geometry to affect the hydraulic potential of our subglacial channel, the ice velocity to advect the channel, and the subglacial water pressure to modulate the shear stress at the ice base using different effective-pressure-dependent sliding laws. We use our model to investigate how these points of coupling can influence ice dynamics, and we examine the implications of the assumption of holding subglacial properties fixed during transient simulations. We find that the coupled ice–hydrology system creates a zone near the grounding line with high effective pressure. We then show that if the hydrology system is not allowed to evolve with the ice, the ice sheet is much less prone to grounding-line retreat, due to retreat into this zone of high effective pressure. In Section 4, we use a simplified model to further illustrate how the high effective-pressure region develops, and how the transition from high to zero effective pressure at the grounding line is coupled with a large increase in channel cross-sectional area. These results clarify the mechanisms underlying the stark differences in ice-sheet retreat between our transient experiments and between experiments done by others using different sliding laws (Brondex et al., 2017). Our simplified model analysis also provides a physical basis for the assumption of full hydrologic connectivity to the ocean for regions near the grounding line. Despite limitations to our approach related to the simplifying assumptions discussed in Section 4, our findings highlight how potentially important actively evolving subglacial hydrological systems could be for marine-terminating ice-sheet retreat.

Our model limitations serve as motivation for future work to incorporate more physics into similar models, such that they can apply to greater range of settings. A first next step is including additional drainage elements, especially since a channel only occupies a limited portion of the bed and the pressure in the channel may not accurately represent pressures across large areas of

the bed. Adding additional subglacial hydrology elements such as a coupled channel-cavity system would better represent the full subglacial hydrology environment and could facilitate resolving seasonal effects (e.g., Pimentel et al., 2010; Kingslake and Ng, 2013; Hewitt, 2013). The subglacial hydrology component can also be expanded to include additional terms representing mechanisms such as the pressure-dependence of the melting point (e.g., Clarke, 2005; Werder, 2016). Another step is including additional points of coupling between the ice and hydrology models, for example, basal frictional melting that is a function

of basal sliding and influences water flux in the drainage system (Hoffman and Price, 2014). Additional areas of investigation could include coupling the model with geothermal heat flux (Smith-Johnsen et al., 2020), tidal forcing at the grounding line (Rosier et al., 2015), or groundwater aquifer flow and deformation (e.g., Li et al., 2022; Robel et al., 2023).

*Code availability.* The code for the model and figures in this manuscript is found here: https://github.com/glugeorge/coupled_ice_hydrology

## Appendix A: Nondimensionalization

In the following the scaling of the hydrology equations broadly follows Fowler (1999) and Kingslake (2013). The scaling of the ice-sheet equation follows a similar approach. From Eqs. (1), (2), (7)-(10), we define the following scales:

$$S = S_0 S', t = t_0 t', t_h = t_{h0} t', m = m_0 m', N = N_0 N', u = u_0 u', x = x_0 x', Q = Q_0 Q',$$

$$M = M_0 M', \psi = \psi_0 \psi', h = h_0 h', b = h_0 b', a = a_0 a'.$$

We use the same scale for $b$ and $h$. Replacing the variables in Eq. (7) with their corresponding scales multiplied by their dimensionless variables from above and setting the first three coefficients equal yields

$$\frac{S_0}{t_{h0}} = \frac{m_0}{\rho_i} = K_0 S_0 N_0^3, \tag{A1}$$

which results in $N_0 = (K_0 t_{h0})^{-1/3}$ and $t_{h0} = \frac{\rho_i S_0}{m_0}$. Setting $\beta = u_0 \frac{t_{h0}}{x_0}$ gives the nondimensional version of Eq. (7):

$$\frac{\partial S'}{\partial t'_h} = m' - S' N'^3 - \beta u' \frac{\partial S'}{\partial x'}. \tag{A2}$$

Replacing the dimensional variables in Eq. (8) gives

$$\frac{S_0}{t_{h0}} \frac{\partial S'}{\partial t'_h} + \frac{Q_0}{x_0} \frac{\partial Q'}{\partial x'} = \frac{m_0}{\rho_i} r m' + M_0 M', \tag{A3}$$

where $r = \rho_i/\rho_w$. Substituting $t_{h0} = \frac{\rho_i S_0}{m_0}$, we obtain

$$\frac{m_0}{\rho_i} \frac{\partial S'}{\partial t'_h} + \frac{Q_0}{x_0} \frac{\partial Q'}{\partial x'} = \frac{m_0}{\rho_i} r m' + M_0 M'. \tag{A4}$$

We define $M_0 = \frac{Q_0}{x_0}$ and the nondimensional parameter $\epsilon \equiv \frac{x_0 m_0}{Q_0 \rho_i}$, yielding the nondimensional form of Eq. (8):

$$\epsilon \frac{\partial S'}{\partial t'_h} + \frac{\partial Q'}{\partial x'} = \epsilon r m' + M'. \tag{A5}$$

We nondimensionalize $\psi$ using Eq. (11), and choose $\psi_0 = \rho_w g \frac{h_0}{x_0}$. Replacing the dimensional variables in Eq. (9) gives

$$\psi_0 \psi' + \frac{N_0}{x_0} \frac{\partial N'}{\partial x'} = f \rho_w g \frac{Q_0^2}{S_0^{8/3}} \frac{Q'|Q'|}{S'^{8/3}}, \tag{A6}$$

which we use to define $S_0 = \left( f \rho_w g \frac{Q_0^2}{\psi_0} \right)^{3/8}$ and $\delta \equiv \frac{N_0}{x_0 \psi_0}$. This gives the nondimensional form of Eq. (9):

$$\psi' + \delta \frac{\partial N}{\partial x} = \frac{Q'|Q'|}{S'^{8/3}}. \tag{A7}$$

Replacing the dimensional variables in Eq. (10) and equating the left side with the first term on the right defines

$$m_0 = \frac{Q_0 \psi_0}{L} \tag{A8}$$

and yields the nondimensional form of Eq. (10):

$$m' = Q' \left( \psi' + \delta \frac{\partial N'}{\partial x'} \right). \tag{A9}$$

Turning to the ice flow equations, replacing the dimensional variables in Eq. (1) gives

$$\frac{h_0}{t_0} \frac{\partial h'}{\partial t'} + \frac{h_0 u_0}{x_0} \frac{\partial (h'u')}{\partial x'} = a_0 a'. \tag{A10}$$

We set $t_0 = \frac{x_0}{u_0}$. A balance of the accumulation flux with ice-flow over the grounding line leads to $a_0 x_0 = u_0 h_0$. Combining these two expressions leads to $a_0 = \frac{h_0}{t_0}$. And these expressions for $t_0$ and $a_0$ lead to the nondimensional version of Eq. (1):

$$\frac{\partial h'}{\partial t'} + \frac{\partial (h'u')}{\partial x'} = a'. \tag{A11}$$

The nondimensionalization of Eq. (2) differs slightly depending on which sliding law we use; the ice velocity scale, $u_0$,
is different in each case. Replacing the dimensional variables in Eq. (2) using the regularized Coulomb sliding law, $\tau_b = C_C N \left( \frac{u}{u + A_s C_C^n N^n} \right)^{1/n}$, gives

$$2\bar{A}^{-1/n} \frac{h_0 u_0^{1/n}}{x_0^{1/n+1}} \frac{\partial}{\partial x'} \left[ h' \left| \frac{\partial u'}{\partial x'} \right|^{1/n-1} \frac{\partial u'}{\partial x'} \right] - C_C N_0 N' \left( \frac{u_0 u'}{u_0 u' + A_s C^n N_0^n N'^n} \right)^{1/n} - \rho_i g \frac{h_0^2}{x_0} h' \frac{\partial (h' - b')}{\partial x'} = 0. \tag{A12}$$

Defining $u_0 = A_s C^n N_0^n$, $\alpha = \frac{2 u_0^{1/n}}{\rho_i g A^{1/n} h_0 x_0^{1/n}}$, $\gamma = \frac{C_C N_0 x_0}{\rho_i g h_0^2}$, the nondimensional version of Eq. (2) is

$$\alpha \frac{\partial}{\partial x'} \left[ h' \left| \frac{\partial u'}{\partial x'} \right|^{1/n-1} \frac{\partial u'}{\partial x'} \right] - \tau_b' - h' \frac{\partial (h' - b')}{\partial x'} = 0, \tag{A13}$$

where $\tau_b' = \gamma N' \left( \frac{u'}{u' + N'^n} \right)^{1/n}$. Alternatively, when using the Budd sliding law, $\tau_b = C_B N^q u^{1/n}$, replacing the dimensional variables in Eq. (2) yields

$$2\bar{A}^{-1/n} \frac{h_0 u_0^{1/n}}{x_0^{1/n+1}} \frac{\partial}{\partial x'} \left[ h' \left| \frac{\partial u'}{\partial x'} \right|^{1/n-1} \frac{\partial u'}{\partial x'} \right] - C_B N_0 N' u_0^{1/n} u'^{1/n} - \rho_i g \frac{h_0^2}{x_0} h' \frac{\partial (h' - b')}{\partial x'} = 0. \tag{A14}$$

Dividing through by $\rho_i g \frac{h_0^2}{x_0}$ and setting $u_0 = \left(\frac{\rho_i g h_0^2}{C_B N_0 x_0}\right)^n$ allows $\alpha$ to remain the same as when using a regularized Coulomb law, so the nondimensional equation remains Eq. (A13), where $\tau_b' = N' u'^{1/n}$.

We have now nondimensionalized all the equations. We assign values to $Q_0$, $x_0$, and $h_0$, from which we determine the remaining scales:

$$S_0 = \left(f \rho_w g \frac{Q_0^2}{\psi_0}\right)^{3/8}, t_0 = \frac{x0}{u0}, t_{h0} = \frac{\rho_i S_0}{m_0}, m_0 = \frac{Q_0 \psi_0}{L}, \psi_0 = \rho_w g \frac{h_0}{x_0},$$

$$N_0 = (K_0 t_{h0})^{-1/3}, u_0 = A_s C^n N_0^n, M_0 = Q_0/x_0, a_0 = \frac{h_0}{t_0}.$$

Additionally, we have five nondimensional parameters:

$$\epsilon \equiv \frac{x_0 m_0}{Q_0 \rho_i}, \delta \equiv \frac{N_0}{x_0 \psi_0}, \beta \equiv t_{h0}/t_0, \alpha \equiv \frac{2 u_0^{1/n}}{\rho_i g A^{1/n} h_0 x_0^{1/n}}, \gamma \equiv \frac{C N_0 x_0}{\rho_i g h_0^2}.$$

These parameter and scales equations are reduced to be in terms of imposed constants in Table 3. To re-arrange the hydrology equations into 3 equations for our solver, we first combine Eqs. (A7) and (A9) to get $m' = \frac{|Q'|^3}{S'^{8/3}}$. Then substituting into Eq. (A2), we get

$$\frac{\partial S'}{\partial t_h'} = \frac{|Q'|^3}{S'^{8/3}} - S' N'^3 - \beta u' \frac{\partial S'}{\partial x'}. \tag{A15}$$

Then, subbing this into Eq. (A5), we get our equation for $\frac{\partial Q'}{\partial x'}$:

$$\frac{\partial Q'}{\partial x'} = \epsilon(r-1)\frac{|Q'|^3}{S'^{8/3}} + \epsilon\left(S' N'^3 + \beta u' \frac{\partial S'}{\partial x'}\right) + M'. \tag{A16}$$

Finally, re-arranging Eq. (A7) for $\frac{\partial N'}{\partial x'}$ provides the third hydrology equation:

$$\frac{\partial N'}{\partial x'} = \frac{1}{\delta}\left(\frac{Q'|Q'|}{S'^{8/3}} - \psi'\right). \tag{A17}$$

## Appendix B: Coordinate Stretching

For the ice flow equations, we use the same coordinate stretching as described in Appendix A in Schoof (2007). We also apply
the same coordinate stretching to the hydrology equations, which are solved on a uniform grid that stretches with the grounding line. Using $\sigma x_g = x$ and $\tau = t_h$, we find that $\frac{\partial}{\partial x}$ and $\frac{\partial}{\partial t_h}$ transform into $\frac{1}{x_g}\frac{\partial}{\partial \sigma}$ and $\frac{\partial}{\partial \tau} - \frac{\sigma}{x_g}\frac{\partial x_g}{\partial \tau}\frac{\partial}{\partial \sigma}$, respectively. Applying these coordinate transformations to the three hydrology equations gives

$$\frac{\partial S'}{\partial \tau} = \frac{|Q'|^3}{S'^{8/3}} - S' N'^3 + \frac{1}{x_g}\left(\sigma\frac{\partial x_g}{\partial \tau} - \beta u'\right)\frac{\partial S'}{\partial \sigma}, \tag{B1}$$

$$\frac{1}{x_g}\frac{\partial Q'}{\partial \sigma} = \epsilon(r-1)\frac{|Q'|^3}{S'^{8/3}} + \epsilon\left(S' N'^3 + \frac{\beta u'}{x_g}\frac{\partial S'}{\partial \sigma}\right) + M', \tag{B2}$$

$$\frac{\partial N'}{\partial \sigma} = \frac{x_g}{\delta}\left(\frac{Q'|Q'|}{S'^{8/3}} - \psi'\right). \tag{B3}$$

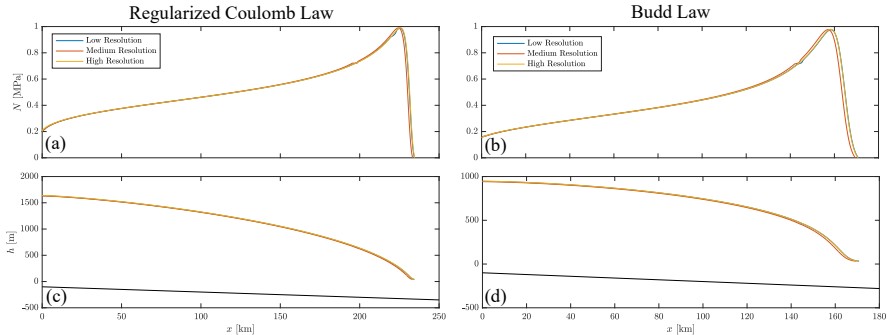

**Figure D1.** The steady-state effective pressure profiles (a,b) and the ice surface height and bed topography (c,d) for experiments S1.B and S1.C, using a range of spatial grids.

## Appendix C: Discretizing hydrology equations

We follow the method described in Appendix A of Schoof (2007) to discretize and solve the ice-flow equations. We follow a similar approach to discretize and solve the hydrology equations. We use centered differences for the spatial derivatives and forward differences for the time derivatives. The discrete equations are as follows:

$$\frac{S_i^j - S_i^{j-1}}{\Delta \tau} = \frac{|Q_i^j|^3}{S_i^{j8/3}} - S_i^j N_i^{j3} + \frac{1}{x_g^j}\left(\sigma_i^j \frac{x_g^j - x_g^{j-1}}{\Delta \tau} - \beta u_i^j\right)\frac{S_{i+1/2}^j - S_{i-1/2}^j}{2\Delta\sigma},\tag{C1}$$

$$\frac{1}{x_g^j}\frac{Q_{i+1/2}^j - Q_{i-1/2}^j}{2\Delta\sigma} = \epsilon(r-1)\frac{|Q_i^j|^3}{S_i^{j8/3}} + \epsilon\left(S_i^j N_i^{j3} + \beta u_i^j \frac{S_{i+1/2}^j - S_{i-1/2}^j}{2x_g^j\Delta\sigma}\right) + M',\tag{C2}$$

$$\frac{\delta}{x_g^j}\frac{N_{i+1/2}^j - N_{i-1/2}^j}{2\Delta\sigma} = \frac{Q_i^j|Q_i^j|}{S_i^{j8/3}} - \psi_i^j,\tag{C3}$$

where $i$ subscripts denote the grid point number and $j$ superscripts denote the time step number.

## Appendix D: Sensitivity to spatial grid and time step size

We perform two additional suites of numerical experiments with both version of the coupled ice-hydrology model (i.e. using both the regularized Coulomb and Budd sliding laws), to ensure that our results do not qualitatively depend on the resolution of spatial grids and time step size.

First, we examine the dependence on the grid resolution by rerunning our S1.B and S1.C experiments with three sets of resolutions: [1] low resolution, with an ice grid with 100 points along 95% of the domain and 200 points along the remaining 5%, and a hydrology grid with 500 points, [2] medium resolution with an ice grid with 100 points along 85% of the domain and 600 points along the remaining 15%, and a hydrology grid of 1000 points, and [3] high resolution grid, with 3000 points for both the hydrology and ice grids. The medium resolution experiment uses the same spatial grids as our sensitivity experiments and the high resolution experiment uses the same grid as S1 and S2 (Table 1).

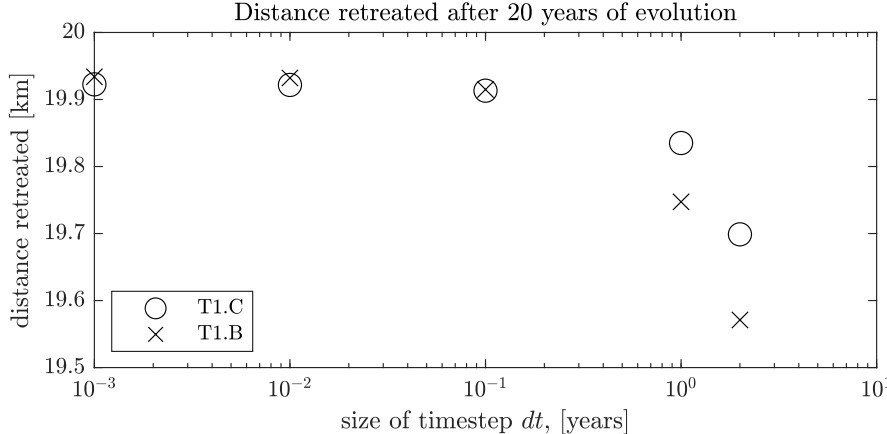

**Figure D2.** Grounding-line retreat after 20 years simulated by the transient, coupled model using a range of time-step sizes and each sliding law. Other than the time steps, all other parameters are the same as used in T1.B and T1.C.

Figure D1 plots steady-state profiles of effective pressure $N$ and the ice-surface height ($h+b$) for experiments [1]-[3]. Across
experiments using both sliding laws, the medium and low resolution profiles closely align with the high resolution profiles. In the medium and low resolution experiments, the maximum effective pressure and the maximum ice thickness agree to the corresponding values in the high resolution experiment to within 0.32% and 0.54%, respectively. The grounding-line position is slightly more sensitive to to grid resolution, varying across the experiments by up to 0.7%. In the low and medium resolution experiments, a minor numerical artifact is visible at the grid junction between the coarse and fine ice grids, likely caused by
the interpolation of variables from the uniform-spaced hydrology grid on to the two different ice grids.

We conclude from these experiments that, grid resolution does not qualitatively impact our main results.

Second, we examine model dependence on time-step size during a crucial part of our transient experiments: the start of the simulations, when the buttressing perturbation is imposed. We perform five additional, short-duration, transient experiments with each sliding law (based on T1.C and T1.B), while varying the time-step size between experiments. Each experiment lasts
20 years and begins with the same perturbation in ice-shelf buttressing used in T1 to trigger retreat. This perturbation is a 10-year-long, linear increase in the buttressing factor $B$ from its initial value of 0.4 to 1. The 20-year simulations therefore cover the 10-year period while $B$ increases and the ice sheet begins to thin and retreat, and the subsequent 10 years of further thinning and retreat. The experiments use a range of time steps: $dt \in \{0.001, 0.01, 0.1, 1, 2\}$ years.

Figure D2 plots the distance retreated by the grounding line at the end of each simulation. Over this wide range of time
step sizes the distance retreated varies by under 2%. Moreover, as the time step size decreases, the distance retreated by the grounding line converges towards to a single value in each set of experiments.

The start of the transient experiments, while the perturbation is being imposed is likely the most sensitive to time-step size. Therefore, the results of this convergence test suggest that the results of our transient experiments do not depend qualitatively on time-step size.

*Author contributions.* GL and JK initiated the study. GL led the modeling and writing, and JK advised on and contributed ideas to the model and discussion in addition to helping to write the manuscript.

*Competing interests.* The authors declare no competing interests.

*Acknowledgements.* The authors acknowledge financial support from the US National Science Foundation's Office of Polar Programs and Columbia University. The NSF award OPP-2003464 that provided primary support for G. Lu's Ph.D. studentship during this study, and that made the Ph.D. studentship possible, results from a proposal written primarily by L. A. Stevens (now at the University of Oxford), with contributions from J. Kingslake and M. Nettles. The authors acknowledge input from and discussions with M. Nettles about this work during its early stages. G. Lu acknowledges the additional financial support of the Natural Sciences and Engineering Research Council of Canada (NSERC), PGS-D 578042. The authors also acknowledge A. Robel and I. Hewitt for separate discussions about modeling, in addition to A. Robel for providing code that forms the foundation of our scripts. The authors acknowledge the three anonymous reviewers for their comments which have improved the manuscript.

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
