# Peer review of "Two-way coupling between ice flow and channelized subglacial drainage enhances modeled marine ice-sheet retreat"

_EGUsphere, 2023_

## Referee Comment (RC2)

**Review of Coupling between ice flow and subglacial hydrology enhances marine ice-sheet retreat**

Anonymous Referee

**1 General Comments**

This work employs a 1D channel subglacial hydrology model to study the effects of dynamically determined effective pressure on ice sheet stability at the grounding line of a flowline ice sheet model. The model employed in this study is a reduced version of the model presented by Kingslake and Ng (2013) (omitting lake evolution and distributed drainage system). The experimental design is an extension of that of Brondex et al. (2017) to include a dynamically determined effective pressure. The authors perform a dimensional analysis on the subglacial hydrology equations to study relative contribution from the involved processes. In the steady state, the authors find that effective pressure reaches a local maximum in the vicinity of the grounding line. The authors state that dynamically determined effective pressure results in faster and farther grounding line retreat than a perscribed effective pressure profile.

This study suffers from three foundational shortcomings: a lack of novelty, an overly simplistic model/experimental design, and non-physical treatment of the static effective pressure in the second transient experiment which forms the foundation for inferring that subglacial hydrology enhances marine ice-sheet retreat. Shoring up these shortcomings will require a non-trivial amount of work but will be a worthwhile contribution for understanding the impact of subglacial hydrology on grounding line positioning.

**2 Specific Comments**

The key finding that effective pressure exhibits a local maximum close to the margin is not itself novel. Others have found an increase in effective pressure toward the ice stream terminus (around 10/13 models, de Fleurian et al., 2018) or grounding line (Dow et al., 2018). It does, however, give the reader more confidence in the model to see it replicate known behaviour.

The inclusion subglacial hydrology to more realistically capturing margin retreat has been done before (e.g. Brondex et al., 2017). The authors state that studies which invert for sliding coefficient on the basis of measured surface velocities underestimate the potential for grounding line retreat and that the inclusion of dynamically determined effective pressure from a subglacial hydrology model reveals enhanced grounding line retreat versus static bed conditions. This conclusion may be a consequence of comparing against a static bed condition which imposes an effective pressure profile which does not change as the ice at a given location thins. This means that the static bed condition case is stabilized by an effective pressure condition from previously thicker ice. These experiments should be redone to reflect ice thinning, for example by imposing an effective

pressure as a fraction of overburden as detailed in the response to Fig 7 below. Furthermore, it should be shown that these findings are robust to varied choices of parameters. For example, how does grounding line retreat differ for the static and dynamic cases for different accumulation rate? This study would benefit from a sensitivity analysis of the conclusions to scale decisions in the experimental design (flow law constant, accumulation rate, additional water source term, and input water flux boundary condition). Or better yet, an internally consistent calculation of some of those quantities.

It is a worthwhile exercise to examine the marginal contribution of isolated processes to overall system behaviour by turning off other interacting processes. In the earth system, however, neglecting fundamental processes and their interactions breaks the relation between model and true system. Here the authors employ a subglacial hydrology model forced with prescribed melt input. In the true earth system, this melt input comes from strain heating, basal frictional heating, and surface melt sources. This model is missing an important set of feedbacks between temperature dependent ice stiffness, basal melt production, hydrologically controlled basal friction, and ice-thermomechanics. Including thermomechanics in this model is feasible given its simplicity and would allow extension of model behaviour closer to the true system.

The perturbation used in the transient experiments is not physical. The time scale for surface temperatures to propagate to the full thickness of the ice body is given by the thickness over the accumulation rate. Given your ice thickness of 1000 m and realistic spatio-temporal mean Holocene accumulation rates for West Antarctica of 0.27 m/a (Bodart et al., 2023) or up to 0.40 m/a modern accumulation rates closer to the coast of West Antarctica (Kaspari et al., 2004), it would take about 2.5 kyr for surface warming to propagate through the ice column – far longer than the 10 yr period examined in the transient experiments. A more realistic approach would be to impose a change in buttressing at the grounding line to simulate ice shelf break up, allowing self-consistent evolution of the ice stiffness with thermomechanics. This would also allow for comparison of the importance of subglacial hydrology to buttressing for Antarctic mass balance and grounding line stability. In the context of the processes which determine grounding line position, where does subglacial hydrology rank?

The summary accumulation rates discussed above also point to an issue with the parameter ranges in the experimental design. No justification is given for those values and in the case of accumulation, the upper bound for the sensitivity experiments is far too high (10 m/a) and the mean value used in the other experiments is too high (1 m/a). Because of the sampling methodology used in the sensitivity analysis, the realistic accumulation rates are undersampled. This is also the region (your Fig. 2) where your model displays the greatest sensitivity to accumulation rate – in terms of effective pressure, ice thickness, and grounding line position. The parameter ranges in this study need to be revisited, adequately justified, and more appropriately sampled (see note below for line 190).

The dimensional analysis in steady state of the model processes was a welcomed inclusion and served to highlight important model components. In all figures other than Fig 5, however, showing the non-dimensionalized versions of the variables made their interpretation cumbersome. These should be changed to the dimensional versions.

**3 Itemized Review**

Numbers refer to line numbers unless otherwise indicated

- 21: Seroussi et al. (2020) does not discuss subglacial hydrology, they only refer to de Fleurian et al. (2018), please remove.

- 27: Also see Alley et al. (1994)

- 33: Drew and Tarasov (2023) shows varied ice sheet behaviour by parameterization and inclusion of varied hydrologies.

- 38: But there are plenty of examples of other subglacial hydrology models of varying degrees of complexity from intermediate (Bueler and van Pelt, 2015; Drew and Tarasov, 2023) to high (Werder et al., 2013; Sommers et al., 2018). I think this sentence could be removed.

- 50: Consider discussing Dow et al. (2018) application of GLADS to marine terminating ice stream but for one-way coupled unsteady conditions. They also find increased effective pressure at grounding line.

- 80: citation needed

- 102: add e.g. to citation

- 167: remove respectively

- 170: a plot of the relative positioning of hydrology and ice dynamic variables and their grids would help here

- 190: How were these parameter ranges decided upon? Please provide justification/citation.

- 190: how was the base vector selected? Did you try multiple base vectors? One at a time sensitivity analyses can be very sensitive to fixed values and is problematic for non-linear coupled models (Saltelli et al., 2019).

- 190: 100 values is quite a lot for a one-at-a-time sensitivity analysis. If able to run the experiment for this density then the sensitivity analysis would benefit from including interaction terms. See for example (Saltelli, 2002). Also, a uniformly spaced sampling is not valid for sampling across several orders of magnitude ($Q_{in}$ varies across 4), the lowest value sampled is actually 0.1 not 0.001 as stated in the text. The plots in fig. 2 show the greatest change at the lower values of each parameter range. Likely the zone of greatest sensitivity has not been sampled and is not displayed (e.g. $A_s$ is stated to be varied between 0.01 and 100 but the lowest value actually sampled would be 1. Log uniform is more appropriate.

- 190: a table of the variables probed and their ranges would help here, perhaps an additional column in Table 2.

- 198: is this hydrology resolution in agreement with table 1?

- table 3: it would be helpful to see the equations in the centre column reduced to the constants in table 2 and the imposed scales in the first 3 rows. This would help to follow the scale choices through to the impact on process.

85    – 235: It looks as though these results are past the CFL limit for an explicit time step length. Taking your hydrology grid resolution as 100 m and the water velocity as $Q_0/S_0 \approx 0.5 m/s$, your time step should be less than 200 s ($\approx 6e - 6$ yr), but in your transient experiments you are using time steps of 5 yr and 0.05 yr.

    You are, however, solving via implicit backward Euler which is stable at larger Courant numbers. How sensitive are these results to your time step length? Do they converge under decreasing time steps? At such large Courant numbers

90     time convergence needs to be checked when dealing with highly-non linear systems.

   – 244: Holding N constant for the T2 experiments neglects any change in coupling due to ice thinning and removes an important positive feedback. Perhaps a more reasonable approach would be to hold N as a fraction of overburden determined by the steady state conditions prior to perturbation? Then you could include that critical positive feedback of ice thinning in your T2 experiments.

95    – Fig. 2: please change horizontal axes to log scale for those parameters which vary over several orders of magnitude. Also changing the distance to actual physical distance instead of normalized distance would help the reader to recall the scales involved.

   – 317: It is not appropriate to call the ice and hydrology fully coupled here, as no melt from basal, englacial, or surficial surfaces is included in a self consistent way here (e.g. strain heating or surface melt due to lowering below the equilibrium

100     line altitude). Also S2 is not coupled to an ice model but forced with a profile (Eq. 19) and constant velocity.

   – 325: It does not follow that the region of higher basal shear stress is caused by the higher effective pressure. If no hydrology were included (e.g. Weerman sliding) you would still get this increase in basal shear stress here because of the higher driving stresses.

   – Fig 7: This grounding line stability in the T2 versus T1 model setups may be a consequence of assuming an effective

105     pressure which is independent of ice sheet thickness. For example if one assumes SIA $- \tau_b = \tau_d -$ and compares the effect on ice velocity from effective pressure proportional to ice sheet thickness ($N = aP_{ice}$) or constant then using your Budd sliding law you get $u = \rho_i g h \frac{\partial h - b}{\partial x}/C_b N_{const}$ versus $u = \frac{\partial h - b}{\partial x}/C_b a$. In the constant effective pressure case the velocity reduction due to ice thinning will be much greater than in the effective pressure proportional to overburden case. Furthermore, as the margin retreats into steadily increasing effective pressure you are effectively reducing the sliding

110     coefficient. In this plot your velocity increase from the ice stiffness perturbation is at least half (plot seems to be cut off) in your T2 scenario versus T1. This disparity should likely be much less and is possibly the reason for the grounding line bifurcation.

   – 359: The driving stress does not decrease because of the lower longitudinal stress increase, driving stress is determined by surface slope. In your Budd sliding equation the basal shear stress increases to balance the driving stress. This means

115     velocity increase and ice thinning. Please restate.

   – 361: Again, this higher N near the boundary is non-physical given thinner ice from retreat.

- paragraph starting at 390: I am confused by the logic here. Applying Eq 21 "which holds for most of the ice sheet far from the grounding line" to conditions right at the grounding line and inferring "that the advection term must grow large near the grounding line." Perhaps the reasoning here can be better expounded in another way or reworded?

- 397: This reasoning feels circular in context of previous paragraph. Also, generally one would expect that as channel cross section gets bigger, flux gets bigger, and effective pressure gets bigger. Effective pressure drops because the ice is thinning toward the margin.

- 399: do you mean $\psi \approx -\frac{\partial N}{\partial x}$?

- 414: which results?

- 404: "also be explained by coupling" – what aspect of the coupling?

- 430: Again, this is may be simply a consequence of the choice of effective pressure profile.

- 435: "holding it static is similar..." – but it is not, when the ice retreats into the region of non-physically higher effective pressure you are effectively reducing the sliding coefficient, not just holding it constant.

**References**

130 Alley, R. B., Anandakrishnan, S., Bentley, C. R., and Lord, N.: A water-piracy hypothesis for the stagnation of Ice Stream C, Antarctica, Annals of Glaciology, 20, 187–194, https://doi.org/10.3189/1994aog20-1-187-194, 1994.

Bodart, J. A., Bingham, R. G., Young, D. A., MacGregor, J. A., Ashmore, D. W., Quartini, E., Hein, A. S., Vaughan, D. G., and Blankenship, D. D.: High mid-Holocene accumulation rates over West Antarctica inferred from a pervasive ice-penetrating radar reflector, The Cryosphere, 17, 1497–1512, https://doi.org/10.5194/tc-17-1497-2023, 2023.

135 Brondex, J., Gagaliardini, O., Gillet-Chaulet, F., and Durand, G.: Sensitivity of grounding line dynamics to the choice of the friction law, Journal of Glaciology, 63, 854–866, https://doi.org/10.1017/jog.2017.51, 2017.

Bueler, E. and van Pelt, W.: Mass-conserving subglacial hydrology in the Parallel Ice Sheet Model version 0.6, Geoscientific Model Development, 8, 1613–1635, https://doi.org/10.5194/gmd-8-1613-2015, 2015.

de Fleurian, B., Werder, M. A., Beyer, S., Brinkerhoff, D. J., Delaney, I., Dow, C. F., Downs, J., Gagliardini, O., Hoffman, M. J., Hooke,
140 R. L., Seguinot, J., and Sommers, A. N.: SHMIP The subglacial hydrology model intercomparison Project, Journal of Glaciology, 64, 897–916, https://doi.org/10.1017/jog.2018.78, 2018.

Dow, C. F., Werder, M. A., Babonis, G., Nowicki, S., Walker, R. T., Csatho, B., and Morlighem, M.: Dynamics of Active Subglacial Lakes in Recovery Ice Stream, Journal of Geophysical Research: Earth Surface, 123, 837–850, https://doi.org/10.1002/2017jf004409, 2018.

Drew, M. and Tarasov, L.: Surging of a Hudson Strait-scale ice stream: subglacial hydrology matters but the process details mostly do not,
145 The Cryosphere, 17, 5391–5415, https://doi.org/10.5194/tc-17-5391-2023, 2023.

Kaspari, S., Mayewski, P. A., Dixon, D. A., Spikes, V. B., Sneed, S. B., Handley, M. J., and Hamilton, G. S.: Climate variability in West Antarctica derived from annual accumulation-rate records from ITASE firn/ice cores, Annals of Glaciology, 39, 585–594, https://doi.org/10.3189/172756404781814447, 2004.

Kingslake, J. and Ng, F.: Modelling the coupling of flood discharge with glacier flow during jökulhlaups, Annals of Glaciology, 54, 25–31,
150 https://doi.org/10.3189/2013aog63a331, 2013.

Saltelli, A.: Making best use of model evaluations to compute sensitivity indices, Computer Physics Communications, 145, 280–297, https://doi.org/10.1016/s0010-4655(02)00280-1, 2002.

Saltelli, A., Aleksankina, K., Becker, W., Fennell, P., Ferretti, F., Holst, N., Li, S., and Wu, Q.: Why so many published sensitivity analyses are false: A systematic review of sensitivity analysis practices, Environmental Modelling & Software, 114, 29–39,
155 https://doi.org/10.1016/j.envsoft.2019.01.012, 2019.

Seroussi, H., Nowicki, S., Payne, A. J., Goelzer, H., Lipscomb, W. H., Abe-Ouchi, A., Agosta, C., Albrecht, T., Asay-Davis, X., Barthel, A., Calov, R., Cullather, R., Dumas, C., Galton-Fenzi, B. K., Gladstone, R., Golledge, N. R., Gregory, J. M., Greve, R., Hattermann, T., Hoffman, M. J., Humbert, A., Huybrechts, P., Jourdain, N. C., Kleiner, T., Larour, E., Leguy, G. R., Lowry, D. P., Little, C. M., Morlighem, M., Pattyn, F., Pelle, T., Price, S. F., Quiquet, A., Reese, R., Schlegel, N.-J., Shepherd, A., Simon, E., Smith, R. S., Straneo, F., Sun, S.,
160 Trusel, L. D., Van Breedam, J., van de Wal, R. S. W., Winkelmann, R., Zhao, C., Zhang, T., and Zwinger, T.: ISMIP6 Antarctica: a multimodel ensemble of the Antarctic ice sheet evolution over the 21st century, The Cryosphere, 14, 3033–3070, https://doi.org/10.5194/tc-14-3033-2020, 2020.

Sommers, A., Rajaram, H., and Morlighem, M.: SHAKTI: Subglacial Hydrology and Kinetic, Transient Interactions v1.0, Geoscientific Model Development, 11, 2955–2974, https://doi.org/10.5194/gmd-11-2955-2018, 2018.

165 Werder, M. A., Hewitt, I. J., Schoof, C. G., and Flowers, G. E.: Modeling channelized and distributed subglacial drainage in two dimensions, Journal of Geophysical Research: Earth Surface, 118, 2140–2158, https://doi.org/10.1002/jgrf.20146, 2013.

---

## Referee Comment (RC3)

Review of: Coupling between ice flow and subglacial hydrology enhances marine ice-sheet retreat by Lu and Kingslake

This is an interesting study that fills a niche by including a representation of subglacial channel drainage in an idealized model of a marine-terminating ice sheet. I have only minor comments for the authors (please see also the annotated pdf) that I hope will help set readers' expectations at the onset, mostly related to (1) the scope of model application including assumed geometry and (2) the rationale for choosing a single channel to describe subglacial drainage morphology. A few additional sentences in the manuscript could also be used to speculate/comment on the extent to which key results in the paper are a function of channelized drainage versus coupled drainage of any type (3).

1. **Scope of application of model:**

   a. Please clarify in the abstract/intro the scope of application of the model. Is it meant to represent a marine ice sheet (i.e., Antarctica) and/or any marine-terminating glacier (e.g., Greenland outlet glaciers and other tidewater glaciers)?

   b. Bed slopes are partly what define, in some conceptions at least, marine ice sheets and tidewater glaciers. A linear bed geometry is mentioned for the first time around Line 195 (unless I missed an earlier mention) and just for a subset of the model runs. Overdeepened bed geometry characteristic of marine ice sheets is not mentioned until after Line 225 (page 20). Establishing the range of geometries used early in the paper will complement the clarification in (a).

   c. Does the model apply to locations where surface melt is an important source to the glacier bed? The governing equations suggest yes, while the scaling (hydrology timescale ~ months) suggests no. Readers could work out whether the prescribed supply term listed in Table 2 is consistent with surface melt, but it would just be easier if the paper made a statement to this effect instead.

2. **Conceptual model of subglacial drainage:**

   a. Please justify adoption of channel-only hydrology approach further than the sentence around Line 60 where channels are said to exist.

   b. The channel is assumed to exist all the way from the ice divide to the grounding line, hence temperate bed assumptions are implicit. How realistic are these requirements for the domains envisioned in Greenland and Antarctica?

   c. Why the choice to employ a single straight channel to represent the entire drainage system when more realistic and only slightly more sophisticated options exist (and have been developed by one of the coauthors)?

   d. Is the pressure-melting term included in channel governing equations. If not, why not? Either way, what are the implications of including/neglecting this term on the simulations with adverse bed slopes?

**3. Results and implications:**

a. A local peak in effective pressure upstream of the grounding line, appearing in both one- and two-way coupled simulations, is a result highlighted in this paper. To what extent is this a reflection of the assumed drainage-system morphology (i.e., a channel), which produces high steady-state effective pressure at high discharge? The authors note that a similar peak in effective pressure was observed in GlaDS (distributed and channelized) simulations in another study, but was this a consequence of channels dominating the drainage in this region? In other words, is a peak in N upstream of the grounding line possible in the absence of channelized/channel-dominated drainage? If not, observed peaks in N upstream of grounding lines might be an interesting basis for inference on the subglacial drainage system, provided other causes (e.g., bed roughness) could be ruled out.

b. Given the key result that a dynamically coupled channel produces grounding line retreat whereas a static treatment of effective pressure does not, I'm left wondering if this is a general feature of introducing coupled hydrology, or whether this is a function of the hydrology being a channel with its characteristic relationship between discharge and effective pressure in steady state. A conceptually simple test would be to change up the drainage morphology (from a channel to a sheet-like/diffusive system), but that seems like an unreasonable amount of extra work for one paper. Instead, perhaps the authors can comment, from their experience modelling both drainage system types, on whether the results would differ qualitatively with a different assumed drainage system morphology.

[revised manuscript text omitted]

---

## Author Comment (AC1)

Reviewer 1:

First of all, sorry for being late for my review

The manuscript presents a suite of experiment with a flowline ice-flow model coupled with a subglacial hydrology model. The transient experiments are very similar to the work of Brondex et al. (2017), where the authors studied the sensitivity of the grounding dynamics to the choice of the friction law, including the Budd and Coulomb friction law, however using a very simplified model, i.e. assuming perfect connectivity to the ocean, for the basal hydrology.

Here, the subglacial hydrology model is for a one-dimensional subglacial channel and follows Fowler (1999) and Drews et al. (2017).

The ice flow and hydrology models at two way coupled and the manuscript present a suite of steady and transient experiments to explore how this coupling could apply dynamics. This is synthetic experiments and the authors acknowledge in the discussion that their hydrology model is fairly simple and that more work would be required to make quantitative comparisons in realistic settings.

They show that the hydrology model leads to a maximum in the effective pressure upstream of the grounding line and that it is essential to have an active model to predict the evolution of the grounding line dynamics.

This manuscript is an original extension of the work of Brondex et al. (2017).

**Thank you very much for your constructive review. Our responses to the comments in grey are in black below. Line numbers refer to the submitted version of the manuscript.**

I only have relatively minor comments, the main point is that the description of the experimental design is relatively complex, and often the differences between some experiments are only on the numerical aspects, i.e. grid resolution or time-steps. For example experiment S1 is part of the sensitivity experiment, the only difference being the grid resolution. It's mainly a matter of presentation, but I have the impression that it would be easier to read by presenting the steady-state experiments a two experiments, a first set to study the sensitivity to the physical parameters and a second experiments using one particular set of physical parameters to study the effect of the coupling.

We agree that we should present the steady-state experiments as two sets, one examining sensitivity, and another using a particular set of parameters to study the effect of the coupling. We will introduce our sensitivity experiments first (around line 188), and explicitly label them as such. We will then specify a selected set of parameters used for the sake of studying the effect of the coupling in further detail, and highlight that as a second set of experiments, after describing the sensitivity experiments (around line 200).

Also while the work of Brondex et al. is referenced in the manuscript, I think it would be also easier to present the transient experiments as an extension of Brondex et al.. The main difference being the forcing used to push the grounding line in the retreat phase. To make

reading easier, I think it could be possible to first summarize the main aim of the experiments before entering the details.

We agree that Brondex et al. (2017) is an important foundation upon which this paper is partially built, but the driving goals behind our transient experiments are different. While Brondex et al. investigates the impact of different sliding laws on retreat, our experiments are focused on the impact of an active hydrology system. We chose to test two sliding laws to demonstrate the ubiquity of the impact that hydrology may have, rather than to test the effect of the two different sliding laws. We will mention Brondex et al. (2017) as a basis for our experimental setup in regards to sliding law choices at line 225.

Detailed comments:

*L25: "Usually, this parameter remains static";* Maybe "usually" is not appropriate as there is more and more applications that try to account for the feedback with the basal hydrology in a more or less parameterized way, as discussed in the next paragraph. So "usually" could maybe be replaced by "often"? or "often, in large scale experiements"?
We will replace "usually" with "often".

Eqs (1) and (2):
these equations are often referred to the "Shallow Shelf" or "Sheflfy Stream" Approximation. Maybe use this notation to avoid confusion?
"b" is not defined
We will refer to those notations and include a definition for b.

L104: *"One drawback associated with this sliding law is that the inclusion of N can result in large, nonphysical stresses »* . I don't understand what you mean by this.
We are reiterating statements from Schoof (2005) which states that equations of this form allow for arbitrarily high shear stresses, regardless of N, which disagrees with Iken's bound, which says that there is an upper limit to stress set by bed geometry. We can see how the "inclusion of N" and "nonphysical stresses" may be unclear. We will instead say how this equation allows for arbitrarily high shear stresses, which is unphysical.

*L172* *"one with coarse resolution and another with fine resolution near the terminus":* Maybe use grounding line instead of terminus? The ice shelf being unbuttressed there is no difference if the grounding line is also the terminus or if there is a shelf.
We will refer to the terminus as the grounding line instead.

*L176*: "through enforcing continuity of thickness at the junction between the two segments of each grid and imposing the flotation condition.". I'm not sure I fully understand this.
We can see how this sentence may be confusing. We will delete "the two segments", and emphasize that the flotation condition is enforced at the grounding line, and not the junction.

*L206*: *"coupled model. S1.B uses";* notation is a bit confusion and at first read one wonders what was S1.A before realizing that B and C refer to Budd and Coulomb

We can see how labeling experiments as S1.B and C suggests the possible existence of S1.A. On first usage, we will introduce the experiments as S1.Budd and S1.Coulomb, referred to as S1.B and S1.C thereafter.

*L209: "the same points of coupling are preserved ». Unclear*
We will expand the sentence to "Note that although different sliding laws are used, the subglacial drainage system and the ice-flow components of the model are coupled in the same way in S1.B and S1.C"

References
Brondex, J., Gagliardini, O., Gillet-Chaulet, F., and Durand, G.: Sensitivity of grounding line dynamics to the choice of the friction law, Journal of Glaciology, 63, 854–866, https://doi.org/10.1017/jog.2017.51, 2017.

Schoof, C.: The effect of cavitation on glacier sliding, Proceedings of the Royal Society A: Mathematical, Physical and Engineering Sciences, 461, 609–627, https://doi.org/10.1098/rspa.2004.1350, 2005

---

## Author Comment (AC2)

Reviewer 2

General Comments

This work employs a 1D channel subglacial hydrology model to study the effects of dynamically determined effective pressure on ice sheet stability at the grounding line of a flowline ice sheet model. The model employed in this study is a reduced version of the model presented by Kingslake and Ng (2013) (omitting lake evolution and distributed drainage system). The experimental design is an extension of that of Brondex et al. (2017) to include a dynamically determined effective pressure. The authors perform a dimensional analysis on the subglacial hydrology equations to study relative contribution from the involved processes. In the steady state, the authors find that effective pressure reaches a local maximum in the vicinity of the grounding line. The authors state that dynamically determined effective pressure results in faster and farther grounding line retreat than a perscribed effective pressure profile.

Thank you very much for the constructive, thorough review. Responding to them has indeed involved a non-trivial amount of work, including redesigning the sensitivity analysis, and rerunning all simulations, but has significantly improved the paper, in our opinion.

This study suffers from three foundational shortcomings: a lack of novelty, an overly simplistic model/experimental design, and non-physical treatment of the static effective pressure in the second transient experiment which forms the foundation for inferring that subglacial hydrology enhances marine ice-sheet retreat. Shoring up these shortcomings will require a non-trivial amount of work but will be a worthwhile contribution for understanding the impact of subglacial hydrology on grounding line positioning.

We recognize some of these shortcomings, and will describe how each will be addressed below. In summary:
- Novelty: as pointed out in the opening paragraph of this review and by the other reviewers, this paper represents an extension of previous work. For example, we take a similar approach as Brondex et al. (2017), but use a more physics-based representation of subglacial hydrology. This representation allows us to examine the physical controls on effective pressure and subsequent grounding-line retreat. This has not been achieved in previous similar studies and we therefore disagree that the study lacks novelty. Most importantly, our work examines a novel set of physical couplings between the flow of ice and the flow of subglacial water, yielding new insights into the physics controlling the coupled system. It also provides a physical basis for a commonly used parameterization of water pressure near the grounding line. We will add a sentence mentioning the latter point to the abstract.
- Simplicity: the model and experiments were carefully designed to balance sophistication/realism (for example, by including longitudinal ice stresses, channel water flow physics, and channel evolution physics) and simplicity (for example, by assuming a single channel, a smooth, idealised bed topography, and a uniform, constant accumulation). This has allowed us to observe new behaviour, such as the distribution of effective pressure and, crucially, to understand the physics controlling that behaviour. If our model or experiments were less simplistic, for example, if they included realistic

topography, a multi-component drainage system, or variable climate forcing, it would be much harder to gain that understanding. We argue that there is space in the literature for idealised reduced models that help us uncover and understand new dynamics that may be at play in the real system, just as there is a place for work using models that attempt to describe a wide range of known processes. The latter cannot be built and improved upon without the former, in our opinion.

- Effective-pressure treatment: One of our transient experiments (T2) keeps the effective pressure constant in time. This appears to be a source of confusion and we aim to clarify our approach below and in the revised manuscript. The reviewer points out this is unphysical. We agree. We impose this unphysical behaviour because it emulates the approach of large-scale ice-sheet modelling. One of our goals with this paper is to highlight the unphysicality of this approach and demonstrate the impact it could have on model behaviour - specifically, reducing simulated grounding-line retreat. When simulating contemporary ice-sheet evolution, spatially varying basal properties are typically determined with an inverse procedure that finds basal properties that allow the simulated surface velocity to best approximate measured surface velocities. These basal properties are then kept constant in time. This is what keeping effective pressure constant in time in experiment T2 emulates. The fact that this approach leads to reduced grounding-line retreat potentially has implications for employing this approach in ice-sheet modelling.

Specific Comments

The key finding that effective pressure exhibits a local maximum close to the margin is not itself novel. Others have found an increase in effective pressure toward the ice stream terminus (around 10/13 models, de Fleurian et al., 2018) or grounding line (Dow et al., 2018). It does, however, give the reader more confidence in the model to see it replicate known behaviour.

We agree that one of our findings supports and elaborates on previous findings that effective pressure exhibits a local maximum close to the margin. We do not claim that we are the first to recognize this. We instead try and elaborate on how this arises. We will refer to these previous findings by others directly in our discussion, around line 416, and emphasize how our results are replicating known behaviour.

The inclusion subglacial hydrology to more realistically capturing margin retreat has been done before (e.g. Brondex et al., 2017). The authors state that studies which invert for sliding coefficient on the basis of measured surface velocities underestimate the potential for grounding line retreat and that the inclusion of dynamically determined effective pressure from a subglacial hydrology model reveals enhanced grounding line retreat versus static bed conditions. This conclusion may be a consequence of comparing against a static bed condition which imposes an effective pressure profile which does not change as the ice at a given location thins. This means that the static bed condition case is stabilized by an effective pressure condition from previously thicker ice. These experiments should be redone to reflect ice thinning, for example by imposing an effective pressure as a fraction of overburden as detailed in the response to Fig 7 below.

Yes, our work builds on that of Brondex et al. (2017) to include a subglacial hydrology component that allows for additional points of coupling. Our inversion for this static bed condition is motivated by existing models which invert for a temporally unvarying basal friction parameter. This is almost like imposing an effective pressure profile which does not change as the ice at a given location thins. You are right that in the static bed condition case, grounding-line migration is stabilized by conditions from previously thicker ice. As explained above, this is something we wanted to examine the implications of, given that it is a widely used approach.

Regarding the suggestion to redo the experiments to "reflect ice thinning", our *coupled* transient experiment (T1) does account for ice thinning because the ice thickness affects the water flow and channel growth (as well as accounting for ice velocity affecting channel growth, and water pressure affecting ice flow and therefore ice thickness). Imposing an effective pressure as a fraction of overburden is a useful idea. Perhaps this can be considered a compromise between the more physics-based T1 and the unphysical (but representative of common ice-sheet modelling practice) T2. We have experimented with variations on this suggestion. For example, a reasonable modification to the suggestion is to scale the effective pressure/overburden ratio ($N/P\_i$) horizontally so that the ratio shrinks or stretches with the model domain as the grounding line retreats or advances. Another approach is to assume full hydraulic connectivity to the ocean (Tsai et al., 2015). This is the approach we compare our model to analytically in the discussion. These, and others, are interesting potential parameterizations to explore and it would be interesting to examine if and under what circumstances they yield similar results to our coupled model. However, we argue that this is beyond the scope of the current paper. Our focus is using a simple comparison between a static-in-time, variable-in-space, bed assumption (that can be considered the first-order approach; a zeroth order approach would be to ignore spatial variability as well) to our physics-based model to understand some of the physics of the coupled system. Our focus is not to examine alternative, more complex effective-pressure parameterizations, though this could be a fascinating follow-on study.

Furthermore, it should be shown that these findings are robust to varied choices of parameters. For example, how does grounding line retreat differ for the static and dynamic cases for different accumulation rate? This study would benefit from a sensitivity analysis of the conclusions to scale decisions in the experimental design (flow law constant, accumulation rate, additional water source term, and input water flux boundary condition). Or better yet, an internally consistent calculation of some of those quantities.

We agree that ideally our transient experiments would be run repeatedly over a range of parameter values. Unfortunately, the computational resources needed for this make it highly challenging to perform a comprehensive parameter sweep. Therefore, we followed previous similar work (e.g., Brondex et al., 2017) in neglecting such a sensitivity experiment of the transient experiments. We do however include a redesigned sensitivity analysis of the steady state findings.

This paper has uncovered some new model behaviour and examined the physics of this in detail. A useful next step will be to see if the details of this behaviour depend on model parameters. We consider it reasonable to include this in a future publication.

Given this, we acknowledge that the title of the original submission could be interpreted as too absolute. We have changed it to "Two-way coupling between ice flow and channelized subglacial drainage enhances modeled marine ice-sheet retreat". It now includes 'modeled' so that it is clear that this is a statement about how our model behaves rather than a more general statement.

It is a worthwhile exercise to examine the marginal contribution of isolated processes to overall system behaviour by turning off other interacting processes. In the earth system, however, neglecting fundamental processes and their interactions breaks the relation between model and true system. Here the authors employ a subglacial hydrology model forced with prescribed melt input. In the true earth system, this melt input comes from strain heating, basal frictional heating, and surface melt sources. This model is missing an important set of feedbacks between temperature dependent ice stiffness, basal melt production, hydrologically controlled basal friction, and ice-thermomechanics. Including thermomechanics in this model is feasible given its simplicity and would allow extension of model behaviour closer to the true system.

We appreciate the acknowledgement of how focusing on isolated processes can be a worthwhile exercise. Including thermomechanics is feasible, but we chose to exclude that from our scope in the interests of simplicity. It is true that our model is simpler than reality. However, choosing whether or not to include additional physics is a compromise between simplicity, which greatly aids interpretability, and realism, which can be useful but is not the only factor. For example, because we exclude melt-sliding-hydrology feedbacks, we are able to attribute the interesting model behaviour we observed to other processes. Without this exclusion, it would be more difficult to do this. One negative outcome of this exclusion is that we need to impose a poorly constrained melt water supply term. However, it is important to note that including ice-bed-frictional melting would entail a similar issue; we would need to prescribe the width of channel catchment in the across-flow direction. Finally, note that we are focused on a set of important couplings that have not been looked at together in this setting, whereas melt-sliding-hydrology couplings have been studied in similar models and settings previously (e.g., Robel et al., 2013).

The perturbation used in the transient experiments is not physical. The time scale for surface temperatures to propagate to the full thickness of the ice body is given by the thickness over the accumulation rate. Given your ice thickness of 1000 m and realistic spatio-temporal mean Holocene accumulation rates for West Antarctica of 0.27 m/a (Bodart et al., 2023) or up to 0.40 m/a modern accumulation rates closer to the coast of West Antarctica (Kaspari et al., 2004), it would take about 2.5 kyr for surface warming to propagate through the ice column – far longer than the 10 yr period examined in the transient experiments. A more realistic approach would be to impose a change in buttressing at the grounding line to simulate ice shelf break up, allowing self-consistent evolution of the ice stiffness with thermomechanics. This would also allow for

comparison of the importance of subglacial hydrology to buttressing for Antarctic mass balance and grounding line stability. In the context of the processes which determine grounding line position, where does subglacial hydrology rank?

We agree that instantaneous temperature change is unrealistic. The motivation for this idealized approach came from Schoof (2007), which used this approach in an idealized model. In the revised manuscript, we instead perturb the ice-shelf buttressing, which is parameterized by a multiplier in the ice-stress boundary condition at the grounding line, consistent with the experiments done by Brondex et al. (2017). This adjustment to the experimental design does not affect our results qualitatively and we will edit the methods and the results to account for this new perturbation.

We think that a significantly more complex experimental design would be required to tackle the interesting question the reviewer poses "In the context of the processes which determine grounding line position, where does subglacial hydrology rank?" but this would definitely be an interesting question to ask in future work.

The summary accumulation rates discussed above also point to an issue with the parameter ranges in the experimental design. No justification is given for those values and in the case of accumulation, the upper bound for the sensitivity experiments is far too high (10 m/a) and the mean value used in the other experiments is too high (1 m/a). Because of the sampling methodology used in the sensitivity analysis, the realistic accumulation rates are undersampled. This is also the region (your Fig. 2) where your model displays the greatest sensitivity to accumulation rate – in terms of effective pressure, ice thickness, and grounding line position. The parameter ranges in this study need to be revisited, adequately justified, and more appropriately sampled (see note below for line 190).

We agree. We have changed our sensitivity tests to use realistic bounds, and to also examine interaction terms. We describe the bounds in detail after your note for line 190, and we have modified our discussion based on these new results. We've also modified our remaining experiments to use realistic accumulation rates.

The dimensional analysis in steady state of the model processes was a welcomed inclusion and served to highlight important model components. In all figures other than Fig 5, however, showing the non-dimensionalized versions of the variables made their interpretation cumbersome. These should be changed to the dimensional versions.

We are pleased that the reviewer found the dimensional analysis useful. We will change Figure 4 (Figure 2 in the revised manuscript) so that it plots dimensional variables. However, in Figures 3 and 4, we do not plan to change the nondimensional distance. It is useful to plot against the nondimensional distance because it allows for easier comparison between different simulations, which in general result in different domain lengths.

Itemized Review
Numbers refer to line numbers unless otherwise indicated

– 21: Seroussi et al. (2020) does not discuss subglacial hydrology, they only refer to de Fleurian et al. (2018), please remove.

We will remove this citation.

– 27: Also see Alley et al. (1994)

This is another example of hydrology affecting ice dynamics, and we will include that in our references.

– 33: Drew and Tarasov (2023) shows varied ice sheet behaviour by parameterization and inclusion of varied hydrologies.

We will include this citation along with Kazmierczak et al. (2022) after the statement about ice sheet dynamics and varied hydrologies

– 38: But there are plenty of examples of other subglacial hydrology models of varying degrees of complexity from intermediate (Bueler and van Pelt, 2015; Drew and Tarasov, 2023) to high (Werder et al., 2013; Sommers et al., 2018). I think this sentence could be removed.

Agreed, we will remove this sentence.

– 50: Consider discussing Dow et al. (2018) application of GLADS to marine terminating ice stream but for one-way coupled unsteady conditions. They also find increased effective pressure at grounding line.

This is a useful paper, and we will discuss it at line 366 along with other findings of increased effective pressure near the grounding line.

– 80: citation needed

We will add a citation for the model (Schoof, 2007).

– 102: add e.g. to citation 70

We will add "e.g." to this citation

– 167: remove respectively

We will remove the word "respectively".

– 170: a plot of the relative positioning of hydrology and ice dynamic variables and their grids would help here

We do not plan on adding another figure here showing the grids. We think that the written description of the grids is sufficient. We are happy to revisit this decision if the editor considers such a figure important.

– 190: How were these parameter ranges decided upon? Please provide justification/citation.

In line with the earlier comments, we have determined more realistic ranges and provided justifications. The sensitivity test choices are updated as follows in our methods:

We conduct a separate sensitivity analysis using each of our two sliding laws. We vary four parameters while using the regularized Coulomb law: the accumulation rate a, the meltwater supply M, the ice stiffness A, and the sliding law coefficient C_C. For experiments with the Budd law, we vary three parameters: a, M, and A. We use four different values for each parameter. We limited the sensitivity analysis to these parameters and four different values per parameter to reduce computational time and to aid in visualization of the results. We selected the parameters that alter alpha and gamma in different ways. For example, raising C_B and raising A both reduce alpha, so we only examine sensitivity to A for simplicity.

We will introduce these new tests in Section 2.5.1, and discuss these new results in an updated Section 3.1.

The sampled values were chosen as follows.
- Accumulation rate, a: we linearly vary between 0.1 and 0.5 m/y to encompass realistic accumulation rates suggested by the reviewer (Bodart et al., 2023; Kaspari et al., 2004).
- Additional meltwater supply, M: we logarithmically vary between 1e-5 and 1e-3 $m^2s^{-1}$. This is obtained using realistic water fluxes at the grounding line, $Q_{Out}$, ranging from 1 $m^3s^{-1}$ to 100 $m^3s^{-1}$ (Dow et al., 2022; Hager et al., 2022) and then dividing by an ice sheet length scale of 100 km.
- Ice stiffness, A: we logarithmically vary between 3.9e-26 and 4.2e-25 to encompass the range of stiffnesses used by Schoof (2007).
- Coulomb sliding law coefficient: we linearly vary between 0.1 and 0.5. This is to encompass values of C used in studies by Helanow et al. (2021) and Pimentel et al. (2010).

We then update our S1 experiments to be run with the median of all the sampled parameters. For our transient experiments, we use a combination of parameters that allows for the initial ice-sheet geometry to extend past the overdeepening in the bed geometry.

– 190: how was the base vector selected? Did you try multiple base vectors? One at a time sensitivity analyses can be very sensitive to fixed values and is problematic for non-linear coupled models (Saltelli et al., 2019).

We no longer conduct our sensitivity analysis in a one-at-a-time manner following this and the below suggestion. We selected the base vector for our high resolution example to correspond to the median value of each parameter.

– 190: 100 values is quite a lot for a one-at-a-time sensitivity analysis. If able to run the experiment for this density then the sensitivity analysis would benefit from including interaction terms. See for example (Saltelli, 2002). Also, a uniformly spaced sampling is not valid for sampling across several orders of magnitude (Qin varies across 4), the lowest value sampled is actually 0.1 not 0.001 as stated in the text. The plots in fig. 2 show the greatest change at the lower values of each parameter range. Likely the zone of greatest sensitivity has not been sampled and is not displayed (e.g. As is stated to be varied between 0.01 and 100 but the lowest value actually sampled would be 1. Log uniform is more appropriate.

We agree. We have updated our sensitivity test and sampling to accommodate these suggestions, as described above. Figure 3 and 4 (now 2 and 3) have been completely redesigned as a result.

We have detailed the variables probed and their ranges as an additional column in Table 2.

This is a typo and has been updated to the table 1 values.

Thanks for the suggestion. We have reduced the equations and updated them in the table.

We agree that capturing fully channelized drainage dynamics would require smaller timesteps that adhere to the CFL limit. However, we emphasize that based on our scaling analysis, we assume that the subglacial channel is in a pseudo-steady state, meaning that the ice thickness and velocity evolves slowly enough that the channel size is always effectively in a steady state. This allows us to solve a steady-state version of the hydrology model in each time step, avoiding the requirement to meet the CFL condition for the water flow. This is now explained at the end of section 2.4, where we say: "We exploit this to simplify the numerical solution of the hydrology equation, by setting the time derivative in Eq. (12) to zero". We also include a convergence test (Appendix D), and we see that the first 20 years of simulation converge to the same grounding line position as the time steps shrink.

As discussed above, the purpose of experiment T2 is not to include critical feedbacks (we agree, it does not). Rather, its purpose is to examine the impact of not including some critical feedbacks. This is motivated by the common practice of inverting for basal properties beneath contemporary ice sheets and keeping those properties constant in time.

– Fig. 2: please change horizontal axes to log scale for those parameters which vary over several orders of magnitude. Also changing the distance to actual physical distance instead of normalized distance would help the reader to recall the scales involved.
This figure has been changed to reflect our multidimensional sensitivity test that also looks at interacting terms, and our new Figure 2 includes logarithmic scales and gives dimensional grounding-line positions.

– 317: It is not appropriate to call the ice and hydrology fully coupled here, as no melt from basal, englacial, or surficial surfaces is included in a self consistent way here (e.g. strain heating or surface melt due to lowering below the equilibrium line altitude). Also S2 is not coupled to an ice model but forced with a profile (Eq. 19) and constant velocity.
We agree; our model neglects that important set of couplings. We will avoid using the term "fully-coupled" when referring to our coupled model, and we will refrain from referring to S2 as coupled. We will instead refer to S2 as a hydrology model with an enforced ice profile and velocity.

– 325: It does not follow that the region of higher basal shear stress is caused by the higher effective pressure. If no hydrology were included (e.g. Weerman sliding) you would still get this increase in basal shear stress here because of the higher driving stresses.
It is true that using a Weertman sliding law would result in high basal shear stress in this location *if* the driving stress were high in this location. However, without a region of high basal shear stress in this location the driving stress would not be higher in this location. In other words, we argue that it is the high basal shear that causes the ice thickness to evolve such that the driving stress becomes high in this location, rather than the other way round. We have left this sentence unchanged.

– Fig 7: This grounding line stability in the T2 versus T1 model setups may be a consequence of assuming an effective pressure which is independent of ice sheet thickness. For example if one assumes SIA – τb = τd – and compares the effect on ice velocity from effective pressure proportional to ice sheet thickness (N = aPice) or constant then using your Budd sliding law you get u = ρigh ∂h−b ∂x /CbNconst versus u = ∂h−b ∂x /Cba. In the constant effective pressure case the velocity reduction due to ice thinning will be much greater than in the effective pressure proportional to overburden case. Furthermore, as the margin retreats into steadily increasing effective pressure you are effectively reducing the sliding coefficient. In this plot your velocity increase from the ice stiffness perturbation is at least half (plot seems to be cut off) in your T2 scenario versus T1. This disparity should likely be much less and is possibly the reason for the grounding line bifurcation.
Yes, we agree with this interpretation. The stability is a result of the assumption that this static effective pressure does not change with the ice thickness and the grounding-line position. As described above, this suggested parameterization of effective pressure may prove useful for describing temporal changes as an alternative to the simplest approach used by many models currently. Future work could investigate this.

– 359: The driving stress does not decrease because of the lower longitudinal stress increase, driving stress is determined by surface slope. In your Budd sliding equation the basal shear stress increases to balance the driving stress. This means velocity increase and ice thinning. Please restate.

Yes, we can see how the way we explained this was confusing. We will rephrase four sentences starting at line 358 as follows: "The removal of buttressing results in a reduction in longitudinal stress and an acceleration in ice flow near the grounding line. In T1, effective pressure is near zero immediately upstream of the grounding line so this acceleration does not lead to sufficiently increased basal shear stress to counteract the reduction in longitudinal stress. The acceleration leads to thinning and grounding-line retreat. In contrast, in Experiment T2, when N does not change with time, grounding-line retreat corresponds to a higher N near the boundary (and therefore higher basal shear stress) which grows to balance the reduction in longitudinal stress stress, preventing further acceleration and retreat."

– 361: Again, this higher N near the boundary is non-physical given thinner ice from retreat.
We agree. See our discussion above about the purpose of emulating the non-physical approach taken by many ice-sheet modelling studies, to examine the consequences of this.

– paragraph starting at 390: I am confused by the logic here. Applying Eq 21 "which holds for most of the ice sheet far from the grounding line" to conditions right at the grounding line and inferring "that the advection term must grow large near the grounding line." Perhaps the reasoning here can be better expounded in another way or reworded?
We can see how this could be confusing. The original text tried to point out why Eq. 21 cannot apply at the grounding line (because it would imply an infinite S). We will reword the argument so that it is made initially only in reference to Eq. 12 and the connection to Eq. 21 will only be made parenthetically and to support the argument that S grows large until the advection term becomes large enough to play a role. The sentence that started on line 392 and the next sentence will be replaced with the following:
"The channel in this region therefore grows until the advection term $\beta u^* \partial u / \partial x$ grows large enough to counteract the melt opening. We note how the fact that Eq. (24) only applies in regions far from the grounding line supports this interpretation; Eq. (24) shows that, for a channel with Q != 0, N = 0 requires S to be infinite. This is not physical, but it is conceptually consistent with S growing large near the grounding line until the advection term, which is neglected in the reduced model, starts to play a role."

– 397: This reasoning feels circular in context of previous paragraph. Also, generally one would expect that as channel cross section gets bigger, flux gets bigger, and effective pressure gets bigger. Effective pressure drops because the ice is thinning toward the margin.
In a sense, the reasoning *is* circular because N and S are self-consistently coupled and in some instances, particularly when we can assume a quasi-steady state, it depends on your perspective which you consider is *causing* the other. To help get this idea across in the paper we will change the first sentence of this paragraph to say "Another way of looking at this is that channel cross-sectional area S growing large at the grounding line facilitates N dropping to zero."

For clarification, we note that in this case the flux is fixed by water-mass conservation, so it is not the case that we expect a higher flux for a larger channel.

– 399: do you mean ψ ≈ −∂N ∂x ?
Yes, this was a typo. It will be corrected.

– 414: which results?
This will be clarified to be "our steady-state experiment results"

– 423: "also be explained by coupling" – what aspect of the coupling?
We will clarify this as follows: "explained by coupling between between ice geometry, basal hydraulic gradient, subglacial-channel size, and effective pressure".

– 430: Again, this is may be simply a consequence of the choice of effective pressure profile.
Yes, it is due to the choice of holding the effective pressure profile static.

– 435: "holding it static is similar..." – but it is not, when the ice retreats into the region of non-physically higher effective pressure you are effectively reducing the sliding coefficient, not just holding it constant.
It is similar because in the case when $C_W$ is held constant in time, its spatial variability also effectively includes spatial variability in N, even if the sliding law used in the inversion does not include the effective pressure.

References

Bodart, J. A., Bingham, R. G., Young, D. A., MacGregor, J. A., Ashmore, D. W., Quartini, E., Hein, A. S., Vaughan, D. G., and Blankenship, D. D.: High mid-Holocene accumulation rates over West Antarctica inferred from a pervasive ice-penetrating radar reflector, The Cryosphere, 17, 1497–1512, https://doi.org/10.5194/tc-17-1497-2023, 2023.

Brondex, J., Gagliardini, O., Gillet-Chaulet, F., and Durand, G.: Sensitivity of grounding line dynamics to the choice of the friction law, Journal of Glaciology, 63, 854–866, https://doi.org/10.1017/jog.2017.51, 2017.

Dow, C. F., Ross, N., Jeofry, H., Siu, K., and Siegert, M. J.: Antarctic basal environment shaped by high-pressure flow through a subglacial river system, Nature Geoscience, 15, 892–898, https://doi.org/10.1038/s41561-022-01059-1, 2022.

Hager, A. O., Hoffman, M. J., Price, S. F., and Schroeder, D. M.: Persistent, extensive channelized drainage modeled beneath Thwaites Glacier, West Antarctica, The Cryosphere, 16, 3575–3599, https://doi.org/10.5194/tc-16-3575-2022, 2022

Helanow, C., Iverson, N. R., Woodard, J. B., and Zoet, L. K.: A slip law for hard-bedded glaciers derived from observed bed topography, Science Advances, 7, eabe7798, https://doi.org/10.1126/sciadv.abe7798, 2021.

Kaspari, S., Mayewski, P. A., Dixon, D. A., Spikes, V. B., Sneed, S. B., Handley, M. J., and Hamilton, G. S.: Climate variability in West Antarctica derived from annual accumulation-rate records from ITASE firn/ice cores, Annals of Glaciology, 39, 585–594, https://doi.org/10.3189/172756404781814447, 2004

Kazmierczak, E., Sun, S., Coulon, V., and Pattyn, F.: Subglacial hydrology modulates basal sliding response of the Antarctic ice sheet to climate forcing, The Cryosphere, 16, 4537–4552, https://doi.org/10.5194/tc-16-4537-2022, 2022.

Pimentel, S., Flowers, G. E., and Schoof, C. G.: A hydrologically coupled higher-order flow-band model of ice dynamics with a Coulomb friction sliding law, Journal of Geophysical Research: Earth Surface, 115, https://doi.org/10.1029/2009JF001621, 2010

Robel, A. A., DeGiuli, E., Schoof, C., and Tziperman, E.: Dynamics of ice stream temporal variability: Modes, scales, and hysteresis, Journal of Geophysical Research: Earth Surface, 118, 925–936, https://doi.org/10.1002/jgrf.20072, 2013.

Schoof, C.: Ice sheet grounding line dynamics: Steady states, stability, and hysteresis, Journal of Geophysical Research: Earth Surface, 112, https://doi.org/10.1029/2006JF000664, 2007

Tsai, V. C., Stewart, A. L., and Thompson, A. F.: Marine ice-sheet profiles and stability under Coulomb basal conditions, Journal of Glaciology, 61, 205–215, https://doi.org/10.3189/2015JoG14J221, 2015.

---

## Author Comment (AC3)

Reviewer 3:

This is an interesting study that fills a niche by including a representation of subglacial channel drainage in an idealized model of a marine-terminating ice sheet. I have only minor comments for the authors (please see also the annotated pdf) that I hope will help set readers' expectations at the onset, mostly related to (1) the scope of model application including assumed geometry and (2) the rationale for choosing a single channel to describe subglacial drainage morphology. A few additional sentences in the manuscript could also be used to speculate/comment on the extent to which key results in the paper are a function of channelized drainage versus coupled drainage of any type (3).

Thank you for this comprehensive and constructive review. Our responses to the comments in grey are in black below.

1. Scope of application of model:
   a. Please clarify in the abstract/intro the scope of application of the model. Is it meant to represent a marine ice sheet (i.e., Antarctica) and/or any marine-terminating glacier (e.g., Greenland outlet glaciers and other tidewater glaciers)?
   It is meant to represent a marine ice sheet, with more relevance to those in Antarctica, which tend to have less seasonal changes in their subglacial hydrology environments. We will specify this in our abstract: "We develop a coupled ice–subglacial-hydrology model to investigate the effects of coupling on the long-term evolution of marine-terminating ice sheets."

   b. Bed slopes are partly what define, in some conceptions at least, marine ice sheets and tidewater glaciers. A linear bed geometry is mentioned for the first time around Line 195 (unless I missed an earlier mention) and just for a subset of the model runs. Overdeepened bed geometry characteristic of marine ice sheets is not mentioned until after Line 225 (page 20). Establishing the range of geometries used early in the paper will complement the clarification in (a).
   We are using the terms "marine ice-sheet" and "marine-terminating" strictly to refer to ice that is grounded below sea level and which flows into the ocean. Under this definition these terms refer equally to prograde and retrograde bed slopes.
   We will add "For b we use either a linear prograde slope or an overdeepened bed (Section 2.5)." when introducing the bed variable b, at line 90.

   c. Does the model apply to locations where surface melt is an important source to the glacier bed? The governing equations suggest yes, while the scaling (hydrology timescale ~ months) suggests no. Readers could work out whether the prescribed supply term listed in Table 2 is consistent with surface melt, but it would just be easier if the paper made a statement to this effect instead.

   Our model parameterized all sources of water with the additional melt term, regardless of if it is from surface melt. Our scaling for the hydrology timescale was determined by the internal dynamics of the channel, so it is not tied with surface melt which motivates a

shorter timescale. In our discussion where we discuss limitations, we will mention how our time scale is derived from channel properties rather than external forcing time scales that may be related to surface melt. At around line 460, we say: "we assumed a pseudo-steady-state in the hydrology component of the model in numerical experiments. This was motivated by a scaling analysis which used the properties of the coupled ice-hydrology system (in particular, $x_0$, $h_0$ and $Q_0$) to derive time scales of the hydrology system, rather than the time scale of external forcings. Therefore, the pseudo-steady-state assumption would not apply if the timescales of, for example, meltwater input to the system were shorter, perhaps due to fluctuations in the flux of surface meltwater reaching the bed."

2. Conceptual model of subglacial drainage:
   a. Please justify adoption of channel-only hydrology approach further than the sentence around Line 60 where channels are said to exist.
      To better explain our motivation for using a channel model only, we will replace the sentence starting on 59 with "For simplicity, we do not model an adjacent distributed drainage system, which would provide meltwater to the channel and influence basal sliding. This approach aids in interpretation of the physics of coupling and is supported by previous modelling suggesting that the pressures in channels and adjacent distributed cavities are closely coupled (Dow et al., 2022; Kingslake and Ng, 2013)."

   b. The channel is assumed to exist all the way from the ice divide to the grounding line, hence temperate bed assumptions are implicit. How realistic are these requirements for the domains envisioned in Greenland and Antarctica?
      This is an unrealistic assumption as at some point away from the grounding line the bed will likely be frozen and/or the discharge would be so small that channels would not persist, so the channel would not exist all the way to the divide. This simplification was for convenience, and we will emphasize that when we describe the channel setup (line 128). Adjusting the model setup so that the channel initiates some distance from the divide would have a similar effect on the results as reducing $Q_{in}$ or M; we examine the sensitivity to M in our sensitivity study. We will add: "We assume that the channel extends to the divide. To avoid this simplification causing the channel to become unrealistically large near the divide, we impose a very small $Q_{in}$."

   c. Why the choice to employ a single straight channel to represent the entire drainage system when more realistic and only slightly more sophisticated options exist (and have been developed by one of the coauthors)?
      The motivation is simplicity. Neglecting a distributed component avoids introducing several poorly constrained parameters and allows us to fully identify and explain the physics underlying the model's behaviour. One disadvantage of this approach is that it requires us to assume the channel effective pressure is representative of the effective pressure of a wide-enough portion of the bed that it impacts basal sliding. We consider this a reasonable compromise, in the interest of simplicity; i.e. including a cavity system would avoid this assumption, but it would make it more complicated to clearly explain the

model's behaviour. We prefer the approach of thoroughly explaining the behaviour of a simpler model that admittedly lacks some potentially important components (just like all models), and then moving forward to include more complexity, which will be pursued in future work.

d. Is the pressure-melting term included in channel governing equations. If not, why not? Either way, what are the implications of including/neglecting this term on the simulations with adverse bed slopes?

It is not included in the model. While the grounding line is on a reverse bed slope, one would expect the inclusion of the pressure melting term to slightly suppress channel size upstream of the grounding line because melt rates would be reduced (or if the pressure gradient were sufficient in magnitude, freeze on would occur). Consequently, one might expect a reduction in channel size whenever the water pressure gradient is high. Such locations can be seen in the middle column of panels in Figure 5, where the yellow curves reach a maximum. Whether or not this reduction in melting (or onset of freeze-on) would make a significant impact on the results is unclear, but future work could include this term and examine this question.

3. Results and implications:

a. A local peak in effective pressure upstream of the grounding line, appearing in both one and two-way coupled simulations, is a result highlighted in this paper. To what extent is this a reflection of the assumed drainage-system morphology (i.e., a channel), which produces high steady-state effective pressure at high discharge? The authors note that a similar peak in effective pressure was observed in GlaDS (distributed and channelized) simulations in another study, but was this a consequence of channels dominating the drainage in this region? In other words, is a peak in N upstream of the grounding line possible in the absence of channelized/channel-dominated drainage? If not, observed peaks in N upstream of grounding lines might be an interesting basis for inference on the subglacial drainage system, provided other causes (e.g., bed roughness) could be ruled out.

Our results cannot answer the important question of whether or not a channel is required for the peak in effective pressure to form. Future work should examine this. However, based on our findings, we hypothesize below that you would indeed see a peak in effective pressure upstream of the grounding line in the case of a distributed cavity system.

A key difference between models of channels and models of distributed cavities is the drainage-system opening term (Eq. 7). In channels this opening is caused by melting, which increases the flux and the total potential gradient, and in cavities opening is additionally from sliding over basal topography, which decreases with effective pressure. A key component of our model solutions that leads to the peak in effective pressure is the large size of the channel near the grounding line. Cavities should also be expected to grow large near the terminus because closure is reduced due to low effective pressure (as it is in channels) and opening is increased due to low effective pressure. Therefore,

because other aspects of channel and cavity dynamics are similar (e.g., flux increasing with drainage-system size and total hydraulic gradient), we expect this to lead to a peak in effective pressure similar to those found in this paper. However, because this is speculative and requires examination with a model that includes cavities, we do not include discussion of this in the paper.

b. Given the key result that a dynamically coupled channel produces grounding line retreat whereas a static treatment of effective pressure does not, I'm left wondering if this is a general feature of introducing coupled hydrology, or whether this is a function of the hydrology being a channel with its characteristic relationship between discharge and effective pressure in steady state. A conceptually simple test would be to change up the drainage morphology (from a channel to a sheet-like/diffusive system), but that seems like an unreasonable amount of extra work for one paper. Instead, perhaps the authors can comment, from their experience modelling both drainage system types, on whether the results would differ qualitatively with a different assumed drainage system morphology.

We agree this is an important next step. We hypothesise on the results in our response immediately above.

Specific comments:
Title: Could title be more precise, e.g., does any coupling to any form of hydrology model enhance retreat? Is two-way coupling required? Is representation of channelized drainage required? I more precise declarative title would be useful.

We have modified our title to "Two-way coupling between ice flow and channelized subglacial drainage enhances modeled marine ice-sheet retreat".

4-5: single channel? channelized system? specify dimension here: 1D profile or 2D plan view?

We will specify the dimensions in the abstract as a 1D profile.

13: This is a little confusing since only steady state hydrological variables were mentioned above.

We will eliminate the reference to steady-state profiles in the abstract and instead just refer to profiles, as the variables described also follow the same profile shapes during the transient experiments.

28: True enough, but these refs relate largely to temperate alpine glaciers or Greenland, whereas this paper focuses on the Antarctic ice sheet by virtue of its focus on marine ice sheets (I think?). Would be preferable to include more relevant examples (Clarke ok).

We will also include Alley et al. (1994) and Stearns et al. (2008) as more relevant sources for Antarctic ice streams.

48: Nice framing of this work in context of what is same as/different than other studies. Might be worth citing Hoffman and Price for early exploration of two-way coupling, though in the H&P case the two-way coupling pertained to the cavity/distributed system instead of a channel.

We have cited Hoffman and Price (2014) here.

51: Ok, but unsatisfying without justification. Please add a few sentences of motivation for this approach (with citations) for the domain of interest. Abstract led me to think the application was Antarctica (marine ice sheet), but intro then refers to marine-terminating glaciers, which made me think of Greenland. Would help to know precisely what the scope of application is: marine ice sheets + marine-terminating ice-sheet outlet glaciers + tidewater glaciers?
We will specify here that we are considering marine-terminating glaciers and ice sheets.

52: None of these three coupling mechanisms seems specific to marine-terminating ice sheets. Tweak wording.
We have changed this sentence to "Our experiments include up to three points of coupling between the ice and hydrology models".

54-55: Not clear to me how geometry would *not* modulate hydraulic gradient. Geometry is required as an input to any hydrology model, so I'm missing something here. I thought this point would be, rather, that the hydraulic gradient evolves with water pressure and is not taken as a static function of ice geometry (as is done in some of Fowler's work).
That is true, the ice geometry evolves with the water pressure, but this is accounted for in the third coupling mentioned in this paragraph. The second coupling is the one that every subglacial hydrology model should require - ice geometry affecting the hydraulic gradient. While it may be obvious, ice geometry being an input to the hydrology model is included here for completeness.

58-59: ...and these convey most of the discharge and/or these play a dominant role in setting the effective pressure in the drainage system at the spatial scales of interest? Something a bit beyond the existence of channels would help motivate the conceptual model of the drainage system adopted here.
Yes, the work from Dow et al. (2022) suggests that the changes in channels influences the surrounding subglacial environment, for regions up to 100 km of either side of the channel. Our response to comment 2a describes the edits we have made to better describe our motivation for this modeling choice.

60: presumably the domains assume entirely temperate conditions
Correct, in response to another comment (2b, above) we now highlight this assumption in the methods section.

63: for cases of both forward and adverse slopes?
We don't look at sensitivity to slopes directly in our steady-state experiments, but our transient experiments indicate that this is the case; as the grounding line retreats through regions with an adverse bed slope, the peak in effective pressure remains. Rather than try to explain this in this paragraph, which is mostly about outlining the structure of the paper, we will defer this discussion to section 3, where we will include an additional sentence saying:
"This peak in N persists throughout the retreat over the prograde and reverse bed slopes."

81: define b here or preferably in (2)
We will define b after Eq. (2).

83: define A bar compared to A
This was a typo - we will consistently use just A.

106: Please note whether N is capped at zero in (5) or (6).
While we don't explicitly enforce a zero-limit on N, N does not become negative in our parameter space. If it were negative, these equations would not apply anymore. We will add "Neither N-dependent sliding laws apply in the case when N<0, so we avoid that scenario in simulations" to clarify this.

114: Curious, given Kinglake's previous work, why the SGD model is restricted to a single channel, vs a channel or channels embedded in a distributed system. At the lateral scales of interest, are the dynamics of a single channel expected to be representative of subglacial drainage? Might help to cast conceptual model in terms of narrow swath of marine ice sheet with a width equal to average channel spacing, if this is the case. It may also be the case that the results are insensitive to (potentially) more realistic assumptions; if this is the case it's worth stating up front in the model description.
See our response to comment 2c above. Additionally, at the end of section 2.2, we will add: "Our one-dimensional ice model can be considered to represent a narrow region of an ice sheet, narrow enough that ice stresses and the effective pressure at the ice-bed interface do not vary in the across-flow direction. Furthermore, we assume that a subglacial channel carved into the ice base exists at the ice-bed interface, is aligned with ice flow, and extends in the along-flow direction from the ice divide to the grounding line. We also assume that the basal effective pressure in this narrow region is equal to the effective pressure in the channel."

Equation 7 and 8: The form of the closure term limits the possible assumed cross-sectional geometries. Worth stating what these are in the model description. Please comment on the decision to neglect the pressure-dependence of the melting temperature, or point out which term includes this effect if not neglected. P(T) seems potentially important for adverse bed slopes such as those characteristic of marine ice sheets and tidewater/fjord-terminating glaciers.
We will specify at line 123: "The closure term in Eq. (7) assumes a circular channel geometry."

121: basal melt only for this application, or is surface melt envisioned?
Surface melt is not envisioned in this case. We have added a statement at line 123: "We assume that M comes directly from the subglacial environment rather than from the ice surface."

127-128: This is interesting because it means the model must assume temperate bed conditions all the way from the ice divide to the grounding line. Might be worth stating this explicitly before and commenting on whether such conditions are common in real-world environments the model is intended to represent.

As mentioned earlier, this may not be realistic as the bed would be frozen further upstream from the grounding line.

131-132: Interesting choice to include this in the model. Presumably the channel is aligned with ice flow and has no tortuosity, hence x in governing equations? Worth stating channel alignment explicitly.

Yes, this assumes that the channel is aligned with ice flow. We will state this explicitly at line 89, as described in our response to the comment at line 114.

138: to be complete, should probably state hydraulic gradient depends on bed gradient too as part of geometry.

Yes, we will add this addition. We will change that sentence to: "The hydrology depends on ice geometry and dynamics. For example, the hydraulic gradient is a function of the bed gradient and the ice overburden pressure gradient. The ice overburden pressure is in turn determined by the ice thickness h, while the velocity…"

We will refer to the dashes as primes instead to avoid confusion.

162: I think this means that seasonal surface melt is not envisioned as an important contributor. Worth stating earlier if true.

Yes, this is based on Dow et al. (2022) which suggests that the high pressure channels are fed primarily by basal melt

185: transient ice-flow only (as implied by scaling) or transient ice flow and drainage?

The transient experiments simulate fully transient ice-flow and pseudo-steady-state hydrology. We will modify this sentence to clarify:

"We conduct a series of steady-state simulations, followed by a series of transient simulations in which the ice evolves transiently and the drainage system evolves, but is assumed to be in a pseudo-steady-state."

Table 1:Some detail in the caption would help readers navigate this table, e.g.: does "fixed" mean constant in time (makes sense for geometry, means steady state for hydrology?), does transience apply to both ice flow and hydrology? Would be nice to clearly indicate in table/caption which tests involve two-way coupling.

We will change 'fixed' to 'constant'. The hydrology evolves in the transient experiments, so in that sense it is transient, it is just in a pseudo-steady-state. We think that changing the title of the right-hand column to explain this distinction would be confusing, particularly when the two more important distinctions are 1) between the overall steady state experiments (S1, S2) and the transient experiments (T1, T2), and 2) between the experiment with constant hydrology but evolving hydrology (T2) and the experiment with transient ice and transient (albeit pseudo-steady-state hydrology).

195-196: I think this is the first mention of bed slope. What does the bed look like in other expts? Is the classical adverse-sloping bed of a marine ice sheet ever considered?
There is an adverse-sloping bed in our transient experiments, but in our steady-state sensitivity tests, we did not use an adverse-sloping bed because, in the absence of buttressing, there are no steady states on an adverse bed slope using the SSA model. The marine-ice sheet bed topography is that from the past experiments (e.g. Schoof, 2007; Brondex et al., 2017).

198: I hope the implications, if any, are discussed later. In the presence of velocity gradients in the refined ice grid, a mismatch between ice and hydrology grids seems potentially problematic for realism of coupling.
This is a great point. We will include an analysis of the dependence of our results on the choice of grids in a new section in the appendix (Appendix D). That analysis shows that when the grid resolution is low (lower than we have used in the rest of the paper) a slight discontinuity in the solutions appears at the junction between the two grids. This numerical artefact is insignificant in solutions that use higher resolutions.

203: Does that mean there is just one bed geometry for all experiments?
We use two bed geometries, one linear slope for the steady-state experiments, and an overdeepened bed for our transient evolution. We will add the following where we define b: "For b we use either a linear prograde slope or an overdeepened bed (Section 2.5)"

Table 2: Why stiffer than for standard temperate ice? Add footnote or note in caption if value is taken from precedent or other work. Would be nice in text to relate this to envisioned source: basal melt only? Otherwise reader needs to convert to some intuitive annual melt rate to make inference about source.
We are using the range of parameters used in the experiments from Schoof (2007), and will include this as a source. We also provide justification for the other ranges used in our sensitivity analysis (see response to reviewer 2 for more details).

225: Why do this instead of perturb, e.g., accumulation, which would be a more realistic cause of thinning/retreat? Takes too long?
We have since modified our perturbation to be more similar to that of Brondex et al. (2017), where we change a buttressing parameter at the ice-ocean boundary condition.

228: Oh! That's good. Please state range of bed shapes used much earlier in the MS.
We will add the following where we define b: "For b we use either a linear prograde slope or an overdeepened bed (Section 2.5)"

Figure 2: Why present Coulomb before Budd? Other parts of text and Fig 5 show Budd then Coulomb. Easier to understand if things come in consistent order.
Our new sensitivity plots show both Budd and Coulomb in one figure.

244-245: This makes sense. I was initially wondering why effective pressure instead of flotation fraction was preserved, given the unphysical possibilities that come with fixed effective pressure.

Thank you.

Figure 4: I always think it is helpful to show the domain geometry. You could do this by plotting h as panel (a) but adding it to the bed slope to that the surface and bed of the domain are shown together at the top of the stack of variables.

Since the grounding line positions are different between these models and we wanted to more closely compare the shapes of these profiles with respect to the grounding-line position, we leave the x-axis nondimensional, and as a result, avoid adding in the bed geometry as the different glacier lengths would create additional offsets in the y-direction. We do show domain geometry in later figures.

271: No boundary conditions needed on S given BC on N at GL and on Q at divide? Asking because steep gradients in N remind me of difficulties with BCs on S near glacier termini.

There is a boundary condition for S at the divide that we neglected to elaborate on. We assume that dS/dx at the divide is 0. However, this has no effect because the ice velocity is also zero at the divide. We will clarify this in the main text with the remaining boundary conditions.

290-291: But they do have points of inflection in these variables just downstream of those in the S1 expts. I don't follow this text as it implies to me that there are no inflection points in N, S in the S2 case.

We are referring to the subtle change in curvature right by the grounding line. We will modify this sentence to be "The main difference is that these variables do not have a point of inflection immediately upstream of the grounding line as seen in the results from the coupled model (S1)."

301-302: Good point. Hadn't thought about this as a reason to include advection, but it makes sense that this has to be a closure mechanism in the absence of freeze-on.

Thank you.

351-353: This is a good example of why it is ~nonsensical to impose temporally fixed beta/N/slipperness distributions in time-dependent simulations.

Yes, we sought to highlight that.

367-369: in discussion of one- vs two-way coupling, it might be worth explicitly reminding the reader that this result is different from that of Hoffman and Price (where two-way coupling made a big difference) because of the drainage system morphology: two way coupling should be really important for cavities/sheet where sliding is the main opening term, but much less important for channels where melt is the opening term.

Our transient experiments highlight the importance of two-way coupling. We have modified the sentence to be:

"Our hydrology-only experiment shows how imposed ice geometries and velocities can produce similar profiles of N, though our transient experiments, discussed later, highlight how two-way

coupling specifically between ice geometry, basal hydraulic gradient, subglacial-channel size, and effective pressure) specifically between ice geometry, basal hydraulic gradient, subglacial-channel size, and effective pressure) is needed for realistic retreat."

471: I still don't understand how this is different than any other hydrology model.
We just mean to say that, to our knowledge, previous work has not coupled together a subglacial channel and an ice sheet using these three couplings. We have shown that these couplings lead to interesting behaviour that has potentially important implications (e.g., S grows large at the grounding line and N peaks upstream of the grounding line due to this set of three couplings and this has implications for, respectively, 1) how we parameterize N at and immediately upstream of the grounding line, and 2) the evolution of ice sheets in models with static bed properties). Therefore, we thought it was worth pointing out that this set of couplings is novel in the opening of the conclusions.

In other words, this sentence is intending to highlight the novelty of the couplings, rather than the novelty of the hydrology model.

References:
Alley, R. B., Anandakrishnan, S., Bentley, C. R., and Lord, N.: A water-piracy hypothesis for the stagnation of Ice Stream C, Antarctica, Annals of Glaciology, 20, 187–194, https://doi.org/10.3189/1994AoG20-1-187-194, 1994.

Brondex, J., Gagliardini, O., Gillet-Chaulet, F., and Durand, G.: Sensitivity of grounding line dynamics to the choice of the friction law, Journal of Glaciology, 63, 854–866, https://doi.org/10.1017/jog.2017.51, 2017.

Dow, C. F., Ross, N., Jeofry, H., Siu, K., and Siegert, M. J.: Antarctic basal environment shaped by high-pressure flow through a subglacial river system, Nature Geoscience, 15, 892–898, https://doi.org/10.1038/s41561-022-01059-1, 2022.

Hoffman, M. and Price, S.: Feedbacks between coupled subglacial hydrology and glacier dynamics, Journal of Geophysical Research: Earth Surface, 119, 414–436, https://doi.org/10.1002/2013JF002943, 2014.

Kingslake, J. and Ng, F.: Modelling the coupling of flood discharge with glacier flow during jökulhlaups, Annals of Glaciology, 54, 25–31, https://doi.org/10.3189/2013AoG63A331, 2013.

Schoof, C.: Ice sheet grounding line dynamics: Steady states, stability, and hysteresis, Journal of Geophysical Research: Earth Surface, 112, https://doi.org/10.1029/2006JF000664, 2007

Stearns, L. A., Smith, B. E., and Hamilton, G. S.: Increased flow speed on a large East Antarctic outlet glacier caused by subglacial floods, Nature Geoscience, 1, 827–831, https://doi.org/10.1038/ngeo356, 2008.

---

## Author Response (AR2)

**Author Response**

Thank you for the additional review and comments. Our responses (in black) to the comments (in gray) are below. References to line numbers refer to the revised manuscript, prior to additional minor revisions.

1. Comment 2d re: pressure melting term. The authors state that "future work could include this term and examine this question" but I think they should also cite past work in the manuscript that has already done that: Werder, M. A. (2016). The hydrology of subglacial overdeepenings: a new supercooling threshold formula. Geophysical Research Letters, 43(5), 2045-2052.
- This work by Werder (2016) is indeed a good reference to cite. We include Werder (2016) along with Clarke (2005) as foundational work on pressure melting that can be used to further our model in the future.
- At line 512, we add: "The subglacial hydrology component can also be expanded to include additional terms representing mechanisms such as the pressure-dependence of the melting point (e.g., Clarke, 2005; Werder, 2016)."

2. Author response "the work from Dow et al. (2022) suggests that the changes in channels influences the surrounding subglacial environment, for regions up to 100 km of either side of the channel." This may be the case, but before using this citation as justification, the authors should examine the extent to which this statement is a function of the model architecture (whereby N or pressure is, by definition, the same in channels as it is in the immediately adjacent sheet, as the latter is bounded by the mesh element edges where channels are defined) and the chosen model parameters. The quantitative value of 100m is surely a function of parameters such as the prescribed sheet width feeding the channels and sheet conductivity?
- Thank you for encouraging further investigation towards this citation. In Dow et al. (2022), they conduct various tests examining the sensitivity of basal water pressure to different poorly constrained parameters, including channel and sheet conductivity. They conclude that the size and distribution of the channels are relatively insensitive to these parameters, though we do note that their Figure 3 indicates that the pressure of the surrounding areas vary by a small percentage. Dow et al. (2022) also acknowledge that there are dependencies on mesh size, but conclude that this is negligible on catchment scales. We believe that the 100 km on either side of the channel approaches that scale. Regardless, in the revised manuscript we are cautious with our claims associated with this citation. We solely use this citation to indicate the plausibility of Antarctic subglacial channels, and that regions immediately adjacent to channels may have pressures that track the channel pressures.
- At line 475, we modify our sentence to re-emphasize that our results are only applicable to regions where the subglacial water pressure is dictated by channelized drainage: "Our findings therefore apply only to regions where channelized drainage systems dictate the water pressure."

**References**

Clarke, G. K.: Subglacial Processes, Annual Review of Earth and Planetary Sciences, 33, 247–276, https://doi.org/10.1146/annurev.earth.33.092203.122621, 2005.

Dow, C. F., Ross, N., Jeofry, H., Siu, K., and Siegert, M. J.: Antarctic basal environment shaped by high-pressure flow through a subglacial river system, Nature Geoscience, 15, 892–898, https://doi.org/10.1038/s41561-022-01059-1, 2022.

Werder, M. A.: The hydrology of subglacial overdeepenings: A new supercooling threshold formula, Geophysical Research Letters, 43, 2045–2052, https://doi.org/10.1002/2015GL067542, 2016.